# The sequences of 150,119 genomes in the UK Biobank

Bjarni V. Halldorsson[1,2 ✉], Hannes P. Eggertsson[1], Kristjan H. S. Moore[1], Hannes Hauswedell[1], Ogmundur Eiriksson[1], Magnus O. Ulfarsson[1,3], Gunnar Palsson[1], Marteinn T. Hardarson[1,2], Asmundur Oddsson[1], Brynjar O. Jensson[1], Snaedis Kristmundsdottir[1,2], Brynja D. Sigurpalsdottir[1,2], Olafur A. Stefansson[1], Doruk Beyter[1], Guillaume Holley[1], Vinicius Tragante[1], Arnaldur Gylfason[1], Pall I. Olason[1], Florian Zink[1], Margret Asgeirsdottir[1], Sverrir T. Sverrisson[1], Brynjar Sigurdsson[1], Sigurjon A. Gudjonsson[1], Gunnar T. Sigurdsson[1], Gisli H. Halldorsson[1], Gardar Sveinbjornsson[1], Kristjan Norland[1], Unnur Styrkarsdottir[1], Droplaug N. Magnusdottir[1], Steinunn Snorradottir[1], Kari Kristinsson[1], Emilia Sobech[1], Helgi Jonsson[4,5], Arni J. Geirsson[4], Isleifur Olafsson[4], Palmi Jonsson[4,5], Ole Birger Pedersen[6], Christian Erikstrup[7,8], Søren Brunak[9], Sisse Rye Ostrowski[10,11], DBDS Genetic Consortium*, Gudmar Thorleifsson[1], Frosti Jonsson[1], Pall Melsted[1,3], Ingileif Jonsdottir[1,5], Thorunn Rafnar[1], Hilma Holm[1], Hreinn Stefansson[1], Jona Saemundsdottir[1], Daniel F. Gudbjartsson[1,3], Olafur T. Magnusson[1], Gisli Masson[1], Unnur Thorsteinsdottir[1,5], Agnar Helgason[1,12], Hakon Jonsson[1], Patrick Sulem[1] & Kari Stefansson[1 ✉]

Detailed knowledge of how diversity in the sequence of the human genome affects phenotypic diversity depends on a comprehensive and reliable characterization of both sequences and phenotypic variation. Over the past decade, insights into this relationship have been obtained from whole-exome sequencing or whole-genome sequencing of large cohorts with rich phenotypic data[1,2]. Here we describe the analysis of whole-genome sequencing of 150,119 individuals from the UK Biobank[3]. This constitutes a set of high-quality variants, including 585,040,410 single-nucleotide polymorphisms, representing 7.0% of all possible human single-nucleotide polymorphisms, and 58,707,036 indels. This large set of variants allows us to characterize selection based on sequence variation within a population through a depletion rank score of windows along the genome. Depletion rank analysis shows that coding exons represent a small fraction of regions in the genome subject to strong sequence conservation. We define three cohorts within the UK Biobank: a large British Irish cohort, a smaller African cohort and a South Asian cohort. A haplotype reference panel is provided that allows reliable imputation of most variants carried by three or more sequenced individuals. We identified 895,055 structural variants and 2,536,688 microsatellites, groups of variants typically excluded from large-scale whole-genome sequencing studies. Using this formidable new resource, we provide several examples of trait associations for rare variants with large effects not found previously through studies based on whole-exome sequencing and/or imputation.

The UK Biobank (UKB)[3] documents phenotypic variation of 500,000 participants across the UK, with a healthy volunteer bias[4]. The UKB whole-genome sequencing (WGS) consortium is sequencing the whole genomes of all the participants to an average depth of at least 23.5×. Here we report on the first data release consisting of a vast set of sequence variants, including single-nucleotide polymorphisms (SNPs), short insertions or deletions (indels), microsatellites and structural variants (SVs), based on WGS of 150,119 individuals. All variant calls were performed jointly across individuals, allowing for consistent comparison of results. The resulting dataset provides an unparalleled opportunity to study sequence diversity in humans and its effect on phenotype variation.

Previous studies of the UKB have produced genome-wide SNP array data[5] and whole-exome sequencing (WES) data[6,7]. Although SNP arrays typically only capture a small fraction of common variants in

[1]deCODE genetics/Amgen Inc., Reykjavik, Iceland. [2]School of Technology, Reykjavik University, Reykjavik, Iceland. [3]School of Engineering and Natural Sciences, University of Iceland, Reykjavik, Iceland. [4]Landspitali-University Hospital, Reykjavik, Iceland. [5]Faculty of Medicine, School of Health Sciences, University of Iceland, Reykjavik, Iceland. [6]Department of Clinical Immunology, Zealand University Hospital, Køge, Denmark. [7]Department of Clinical Medicine, Aarhus University, Aarhus, Denmark. [8]Department of Clinical Immunology, Aarhus University Hospital, Aarhus, Denmark. [9]Novo Nordisk Foundation Center for Protein Research, Faculty of Health and Medical Sciences, University of Copenhagen, Copenhagen, Denmark. [10]Department of Clinical Immunology, Copenhagen University Hospital (Rigshospitalet), Copenhagen, Denmark. [11]Department of Clinical Medicine, Faculty of Health and Clinical Sciences, Copenhagen University, Copenhagen, Denmark. [12]Department of Anthropology, University of Iceland, Reykjavik, Iceland. *A list of authors and their affiliations appears at the end of the paper. ✉e-mail: bjarnih@decode.is; kstefans@decode.is

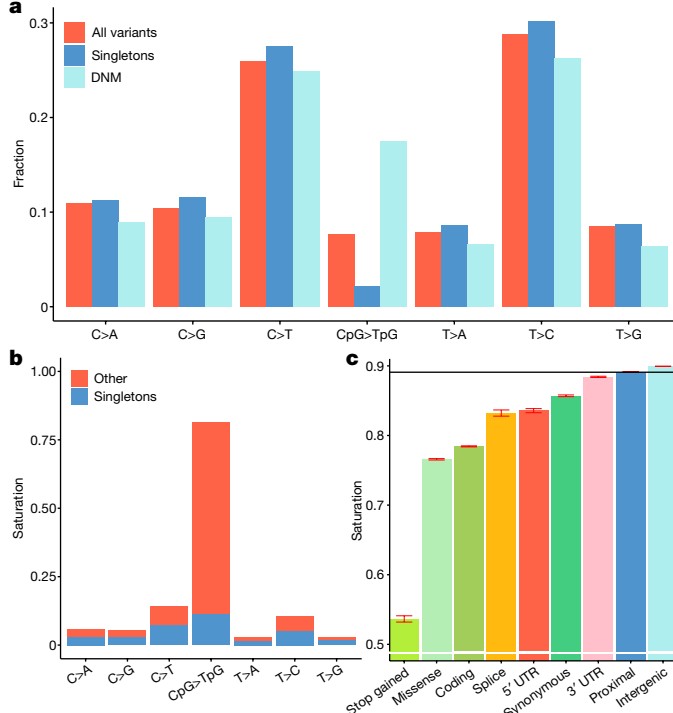

**Fig. 1 | Mutation classes of sequence variants in the UKB. a**, Fraction of SNPs in each mutation class, for all SNPs in our dataset, singletons in our dataset and in an Icelandic set of de novo mutations (DNMs). **b**, Saturation levels of mutations in each class, split into singleton variants (blue) and more common variants (red). **c**, Saturation levels of transitions at methylated CpG sites across genomic annotations and predicted consequence categories. The horizontal line is the average across all methylated CpG sites. The error bars are 95% CIs, which were computed using a normal approximation, treating each CpG site as an independent observation The number of CpG sites used in **c** are: stop gained $n = 46{,}670$, missense $n = 669{,}526$, coding $n = 1{,}067{,}847$, splice $n = 26{,}797$, 5′ UTR $n = 60{,}885$, 3′ UTR $n = 508{,}981$, proximal $n = 17{,}722{,}875$ and intergenic $n = 15{,}266{,}391$.

the genome, when combined with a reference panel of WGS individuals[8], a much larger set of variants in these individuals can be surveyed through imputation. Imputation, however, misses variants private to the individuals only typed on SNP arrays and provides unreliable results for variants with insufficient haplotype sharing between carriers in the reference and imputation sets. Poorly imputed variants are typically rare, highly mutable or in genomic regions with complicated haplotype structure, often due to structural variation.

WES is mainly limited to regions known to be translated and consequently reveals only a small proportion (2–3%) of sequence variation in the human genome. It is relatively straightforward to assign function to variants inside protein-coding regions, but there is abundant evidence that variants outside coding exons are also functionally important[9], explaining a large fraction of the heritability of traits[10].

Large-scale sequencing efforts have typically focused on identifying SNPs and short indels. Although these are the most abundant types of variants in the human genome, other types, including SVs and microsatellites, affect a greater number of base pairs each and consequently are more likely to have a functional impact[11,12]. Even the SVs that overlap exons are difficult to ascertain with WES owing to the much greater variability in the depth of sequence coverage in WES studies than in WGS studies becasue of the capture step of targeted sequencing. Microsatellites, polymorphic tandem repeats of 1–6 bp, are also commonly not examined in large-scale sequence analysis studies.

Here we highlight some of the insights gained from this vast new resource of WGS data that would be challenging or impossible to ascertain from WES and SNP array datasets.

## SNPs and indels

The whole genomes of 150,119 UKB participants were sequenced to an average coverage of 32.5× (at least 23.5× per individual; Supplementary Fig. 1) using Illumina NovaSeq sequencing machines at deCODE Genetics (90,667 individuals) and the Wellcome Trust Sanger Institute (59,452 individuals). Individuals were pseudorandomly selected from the set of UKB participants and divided between the two sequencing centres. All 150,119 individuals were used in variant discovery, 13 individuals were sequenced in duplicate, 11 individuals withdrew consent from time of sequencing to time of analysis and microarray data were not available to us for 135 individuals, leaving 149,960 individuals for subsequent analysis.

Sequence reads were mapped to human reference genome GRCh38[13] using BWA[14]. SNPs and short indels were jointly called over all individuals using both GraphTyper[15] and GATK HaplotypeCaller[16], resulting in 655,928,639 and 710,913,648 variants, respectively. We used several approaches to compare the accuracy of the two variant callers, including comparison to curated datasets[17] (Supplementary Table 1 and Supplementary Fig. 2), transmission of alleles in trios (Supplementary Tables 2 and 3), comparison of imputation accuracy (Supplementary Table 4) and comparison to WES data (Supplementary Table 5). These comparisons suggested that GraphTyper provided more accurate genotype calls. For example, despite 7.7% fewer GraphTyper variants, we estimated that GraphTyper called 4.5% more true-positive variants in trios and had 9.4% more reliably imputing variants than GATK. We therefore restricted subsequent analyses of short variants to the

## Table 1 | Overlap of WES and WGS data

| Annotation | WGS | WES | Intersection of WGS and WES | Unique to WES | Present WES (%) | Missing WES (%) | Present WGS (%) | Missing WGS (%) |
|---|---|---|---|---|---|---|---|---|
| Coding | 6,380,795 | 5,781,829 | 5,686,934 | 94,895 | 89.29 | 10.71 | 98.53 | 1.47 |
| Splice | 445,499 | 397,226 | 388,961 | 8,265 | 87.54 | 12.46 | 98.18 | 1.82 |
| 5′ UTR | 2,125,413 | 590,484 | 572,996 | 17,488 | 27.56 | 72.44 | 99.18 | 0.82 |
| 3′ UTR | 7,214,427 | 764,864 | 743,790 | 21,074 | 10.57 | 89.43 | 99.71 | 0.29 |
| Proximal | 249,702,570 | 6,189,465 | 5,952,145 | 237,320 | 2.48 | 97.52 | 99.91 | 0.09 |
| Intergenic | 292,259,782 | 91,836 | 83,360 | 8,476 | 0.03 | 99.97 | More than 99.99 | Less than 0.01 |

Results are computed for the 109,618 samples present in both datasets and are limited to those variants that are present in at least one individual in either dataset. Numbers refer to the number of variants found in the dataset. WGS refers to the GraphTyperHQ dataset and WES refers to a set of 200,000 WES-sequenced indivdiduals[59]. Missing and present percentages are computed from the number of variants in the union of the two datasets.

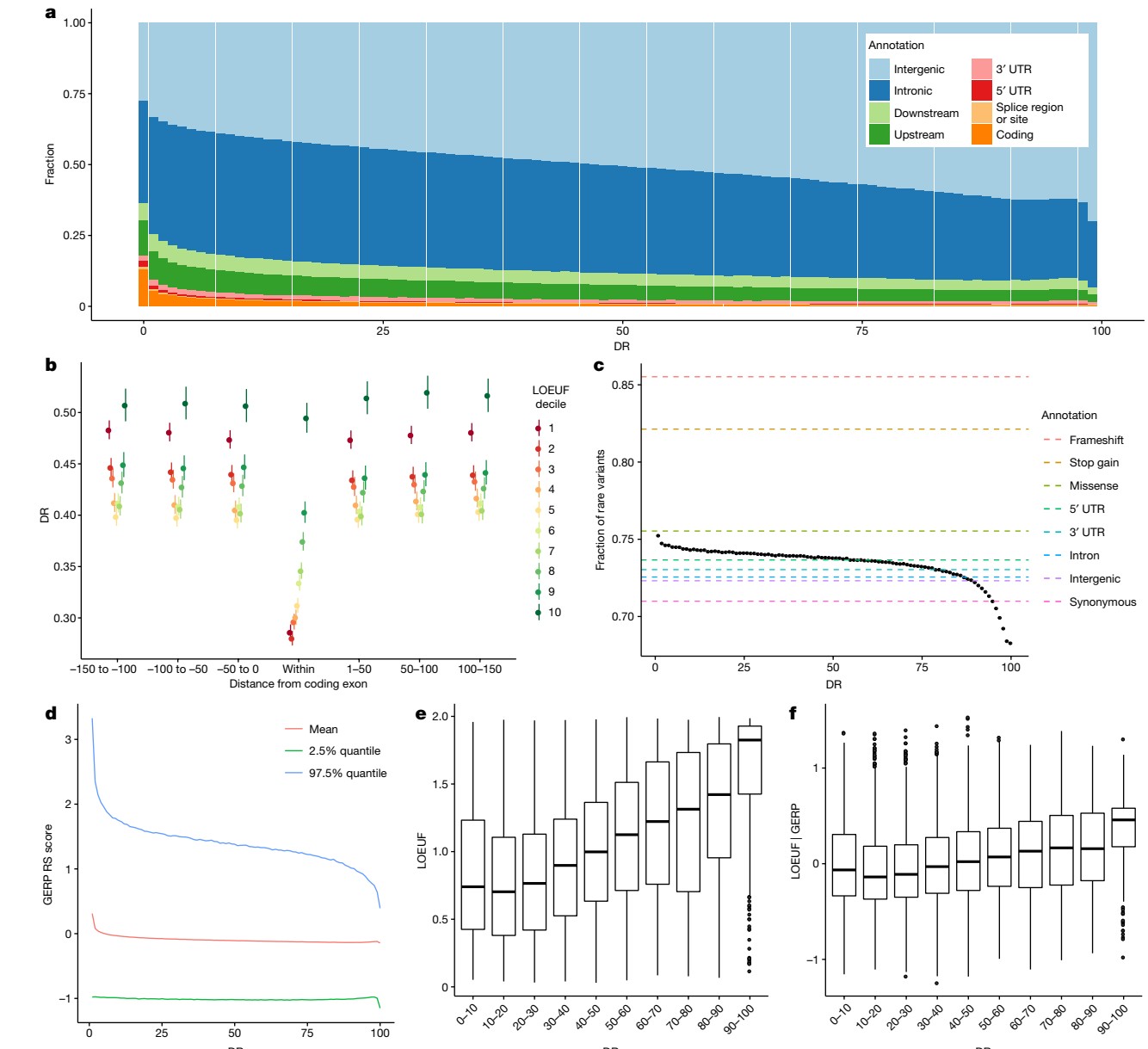

**Fig. 2 | Functionally important regions. a**, Fraction of regions falling into functional annotation classes, as defined by Ensembl gene map, as a function of DR. **b**, DR score as a function of distance from exon and LOEUF decile. Error bars represent 95% CI, computed using a normal approximation, treating each gene (*n* ranges between 1,206 and 1,848) as an independent observation. **c**, Fraction of rare (with four or fewer carriers) variants as a function of DR. **d**, Average GERP score in 500-bp windows as a function of DR. RS, rejected substitution. **e**,**f**, LOUEF (**e**) and LOEUF|GERP (**f**) as a function of DR. In **e** and **f**, middle bar indicates the average, hinges are the 25th and the 75th quantiles, black dots indicate outliers, and the whiskers extend to 1.5 interquartile range from the hinges to the largest or smallest value. The number of genes or observations in the DR ranges are the following: $n_{(0–1)} = 1,234$, $n_{(0.1–0.2)} = 3,202$, $n_{(0.2–0.3)} = 4,474$, $n_{(0.3–0.4)} = 3,888$, $n_{(0.4–0.5)} = 2,476$, $n_{(0.5–0.6)} = 1,384$, $n_{(0.6–0.7)} = 863$, $n_{(0.7–0.8)} = 522$, $n_{(0.8–0.9)} = 374$ and $n_{(0.9–1)} = 427$.

GraphTyper genotypes, although further insights might be gained from exploring these call sets jointly. To contain the number of false positives, GraphTyper uses a logistic regression model that assigns each variant a score (AAscore), predicting the probability that it is a true positive. We focused on the 643,747,446 (98.14%) high-quality GraphTyper variants, indicated by an AAscore above 0.5, hereafter referred to as GraphTyperHQ.

The American College of Medical Genetics and Genomics (ACMG) recommends reporting actionable genotypes in a list of genes associated with diseases that are highly penetrant and for which a well-established intervention is available[18]. We found that 4.1% of the 149,960 individuals carry an actionable genotype in one of 73 genes according to ACMG[18] v3.0. Using WES[6] and ACMG v2.0 (59 genes), 2.0% were reported to

carry an actionable genotype, when restricting our analysis to ACMG v2.0 and the same criteria, we found 2.5% based on WGS, increasing the number of actionable genotypes detected in a large cohort, to the extent that it could have a notable effect on societal disease burden.

The number of variants identified per individual is 40 times larger than the number of variants identified through the WES studies of the same UKB individuals (Table 1; Methods). Although referred to as 'WES', we found that WES primarily captures coding exons and misses most variants in exons that are transcribed but not translated, missing 72.2% and 89.4%, of the 5′ and 3′ untranslated region (UTR) variants, respectively. Even inside of coding exons currently curated by ENCODE[19], we estimate that 10.7% of variants are missed by WES (Table 1). Manual inspection of the missing variants in WES suggests

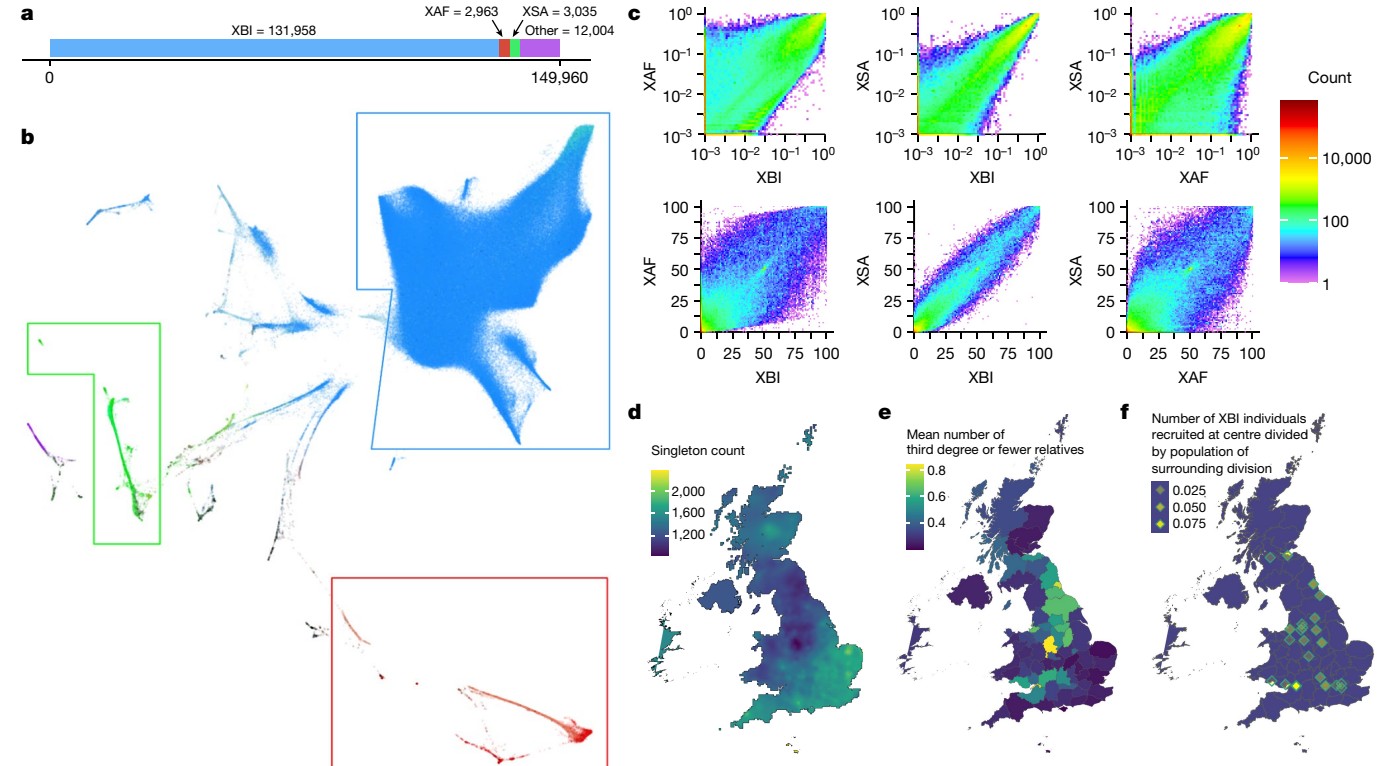

**Fig. 3 | Cohort characteristics. a**, The number of WGS samples analysed for phenotypes in our study. **b**, UMAP plot generated from the first 40 principal components of all UKB participants, coloured by self-reported ethnicity: blue shades for ethnic labels under the white category (XBI), red shades for Black individuals (XAF) and green shades for South Asian individuals (XSA); for the full colour legend, see Supplementary Fig. 17. **c**, Joint frequency spectrum of variants on chromosome 20 between all pairs of populations. **d–f**, Characteristics of the XBI cohort across Great Britain and Ireland are shown: the number of singletons carried by individuals in the XBI cohort as a function of place of birth (**d**); the mean number of third-degree relatives by administrative division (**e**); and the location of UKB assessment centres and estimated fraction of the surrounding population recruited to the UKB (**f**). Differences in singleton counts and the number of third-degree relatives are probably a result of denser sampling of individuals living near UKB assessment centres. Fig. 3d–f by K.H.S.M.

that these are missing due to both missing coverage in some regions and genotyping filters. Conversely, almost all variants identified with WES are found by WGS (Table 1).

## Functionally important regions

The number of SNPs discovered in our study corresponds to an average of one every 4.8 bp, in the regions of the genome that are mappable with short sequence reads. This amounts to detection of 7.0% of all theoretically possible SNPs in these regions (a measure of saturation). We observed 81.5% of all possible autosomal CpG>TpG variants, 11.8% of other transitions and only 4.0% of transversions (Supplementary Table 6). Restricting the analysis to 17,345,777 autosomal CpG dinucleotides methylated in the germ line[9], we observed transition variants at

**Table 2 | Overrepresentation and underrepresentation of GWAS variants in low and high DR regions**

| DR of non-coding regions (%) | Enrichment | 95% CI | *P* value |
|---|---|---|---|
| 1 | 3.22 | 2.44–4.07 | <0.0004 |
| 99 | 0.45 | 0.23–0.70 | <0.0004 |
| 5 | 2.25 | 1.86–2.69 | <0.0004 |
| 95 | 0.61 | 0.47–0.70 | <0.0004 |

Windows overlapping coding exons were removed. Lower DR scores indicate greater sequence conservation.

89.1% of all methylated CpGs. As CpG mutations are so heavily saturated (Fig. 1), the ratio of transitions to transversions (1.66) is lower than found in smaller WGS sets[1] and de novo mutation studies[20].

The vast majority of all variants identified are rare (Supplementary Table 7), 46.0% and 40.6% of all SNPs and short indels, respectively, are singletons (carried by a single sequenced individual), and 96.6% and 91.7% have a frequency below 0.1%. Inference of haplotypes and imputation typically involves identifying variants that are shared due to a common ancestor (are identical by descent). Owing to the scale of the UKB WGS data, an observation of the same allele in unrelated individuals does not always imply identity by descent. A clear indication of this is that only 14% of the highly saturated CpG>TpG variants are singletons, in contrast to 47% for other SNPs (Fig. 1b). These recurrence phenomena have been described in other sample sets using sharing of rare variants between different subsets[2,21]. We used a de novo mutation set from 2,976 trios in Iceland[20] to assess recurrence directly, as variants present in both that set and the UKB must be derived from at least two mutational events. Out of the 194,687 Icelandic de novo mutations, we found 53,859 (27.7%) in the UKB set, providing a direct observation of sequence variants derived from at least two mutational events. As expected, we found that CpG>TpG mutations are the most enriched mutation class in the overlap, owing to their high mutation rate[22] and saturation in the UKB set (Fig. 1b).

The rate and pattern of variants in the genome is informative about the mutation and selection processes that have shaped the genome[23]. The number of sequence variants in the exome has been used to rank genes according to their tolerance of loss of function (LoF) and missense

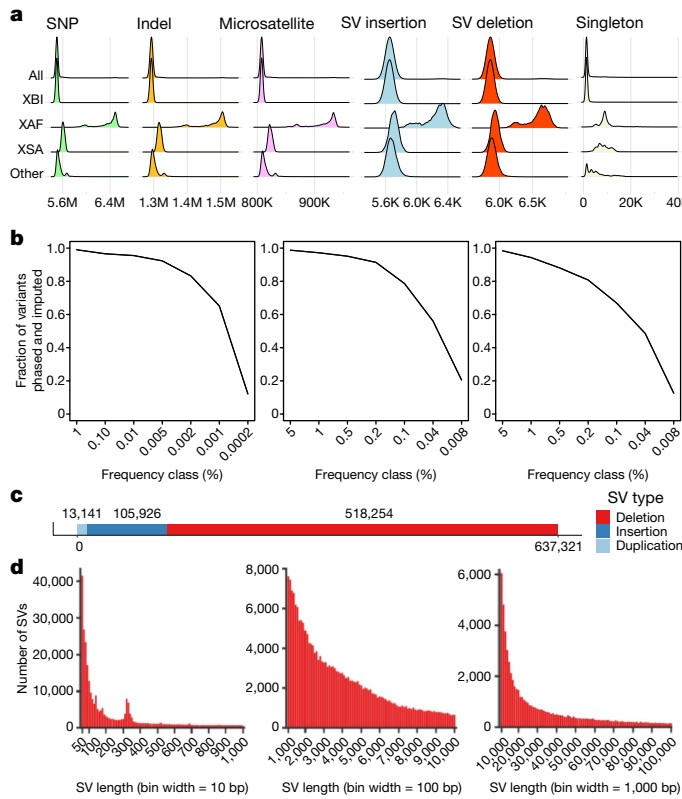

**Fig. 4 | Variant call set. a**, Number of SNPs, indels, microsatellites, SV insertions, SV deletions and singleton SNPs carried per diploid genome of individuals in the overall set and partitioned by population. **b**, Imputation accuracy in the three populations: XBI (left), XAF (middle) and XSA (right). A variant was considered imputed if 'leave one out $r^2$' of phasing was greater than 0.5 and imputation information was greater than 0.8. The $x$ axis splits variants into frequency classes based on the number of carriers in the sequence dataset. Variants are split by variant type. **c**, Number of SVs discovered in the dataset by variant type. **d**, Length distribution of SVs, from 50 to 1,000 bp, 1,000 to 10,000 bp and 10,000 to 100,000 bp.

variation[21,24]. The focus on the exome is because of the availability of WES datasets and the relatively straightforward functional interpretation of coding variants. Conservation across a broad range of species[25] is used to infer the impact of selection beyond the exome, leveraging the extensive accumulation of mutations over millions of years. However, such statistics are only partially informative about sequence conservation specific to humans[26]. Sequence variation in humans[27,28] can be used to characterize human-specific conservation, but large sample sizes are required for accurate inference, as much fewer mutations separate pairs of humans than different species.

The extensive saturation of CpG>TpG variants at methylated CpGs in large WES cohorts has been used to identify genomic annotation or loci where their absence could be indicative of negative selection[21,29]. In line with previous reports[21], we saw less saturation of stop-gain CpG>TpG variants than those that are synonymous (Fig. 1c). Synonymous mutations are often assumed to be unaffected by selection (neutral)[29]; however, we found that synonymous CpG>TpG mutations are less saturated (85.7%) than those that are intergenic (89.9%), supporting the hypothesis that human codon usage is constrained[30].

Extending this approach, we used sequence variant counts in the UKB to seek conserved regions in 500-bp windows across the human genome. We build on the methodology behind the context-dependent tolerance score (CDTS)[27], applying it to a larger dataset. More specifically, we tabulated the number of variants in each window and compared this number to an expected number given the heptamer

nucleotide composition of the window and the fraction of heptamers with a sequence variant across the genome and their mutational classes. We then assigned a rank (depletion rank (DR)) from 0 (most depletion) to 100 (least depletion) for each 500-bp window. As expected, coding exons have a low DR (mean DR = 28.4), but a large number of non-coding regions show even lower DR (more depletion), including non-coding regulatory elements. Among the 1% of regions with the lowest DR, 13.0% are coding and 87.0% are non-coding, with an over-representation of splice, UTR, gene upstream and downstream regions (Fig. 2a). DR increases with distance from coding exons (Fig. 2b). After removing coding exons, among the 1% of regions with lowest and highest DR score, we saw a 3.2-fold and 0.4-fold overrepresentation of GWAS variants, respectively (Table 2), suggesting that the DR score could be a useful prior in GWAS analysis[31]. ENCODE[9] candidate *cis*-regulatory elements are more likely than expected by chance to be found in depleted (low DR) regions (Table 3). Of note, candidate *cis*-regulatory elements located in close proximity to transcription start sites, that is, proximal enhancer-like and promoter-like sequences, are more enriched among depleted regions than distal enhancer-like sequences.

Regions under strong negative selection are expected to have a greater fraction of rare variants (FRV; defined here as variants carried by at most four WGS individuals) than the rest of the genome[28]. We observed a greater FRV in the most depleted regions (DR < 5) than in the least depleted regions (DR > 95): 74.8% versus 69.1% (Fig. 2c and Supplementary Fig. 3). This was also seen when limiting to only non-coding regions (74.6% versus 69.2%). Using the FRV of annotated coding variants as a reference (Fig. 2c), we found that the most depleted regions (DR < 1) had a FRV comparable to missense mutations (75.5%).

Overall, there is a weak correlation between DR and interspecies conservation as measured by genomic evolutionary rate profiling (GERP)[25] (linear regression $r^2 = 0.0050$, two-sided $P < 2.2 \times 10^{-308}$; Fig. 2d). We found a stronger correlation between DR and GERP within coding exons (linear regression $r^2 = 0.0498$, two-sided $P < 2.2 \times 10^{-308}$) than outside them (linear regression $r^2 = 0.0012$, two-sided $P < 2.2 \times 10^{-308}$), indicating that the correlation between DR and GERP is mostly due to the most highly conserved elements, such as coding exons, in the 36 mammalian species used to calculate GERP, with much weaker correlation in less conserved regions.

To determine whether DR reflects human-specific negative selection that is not captured by GERP, we aggregated DR across the exons and compared it to the LOEUF metric from Gnomad[21] (Fig. 2e). LOEUF measures the intolerance to LoF mutations of genes, but it does not measure intolerance outside coding exons. We found that DR is correlated with LOEUF (linear regression $r^2 = 0.085$, two-sided $P < 2.2 \times 10^{-16}$). LOEUF correlates with genes demonstrating autosomal dominant inheritance[21]; in line with this, we found that DR is correlated (linear regression $r^2 = 0.0027$, two-sided $P = 6.6 \times 10^{-12}$) with autosomal dominant genes as reported by OMIM[32] (Supplementary Table 8). Modelling the LOEUF metric as a function of GERP and extracting the residuals from a linear fit, we obtained a measure of human-specific LoF intolerance (LOEUF|GERP). We found that DR is correlated with LOEUF|GERP (linear regression $r^2 = 0.024$, two-sided $P < 2.2 \times 10^{-16}$; Fig. 2f), indicating that DR measures human-specific sequence constraint not captured by GERP. We compared DR with CDTS[27], which is a measure of sequence constraint analogous to the one presented here, and CADD[33], Eigen[34] and LINSIGHT[35], which are measures of functional impact that incorporate interspecies conservation (Extended Data Fig. 1). The constraint metrics that use interspecies conservation form one correlation block (GERP, CADD, Eigen and LINSIGHT) that is less correlated with the DR and CDTS correlation block (Supplementary Table 9). The regions with the lowest DR score show similar enrichment across all metrics (Extended Data Fig. 1). Overall, our results show that DR can be used to help identify genomic regions under constraint across the entire genome and as such provides a valuable resource for identifying non-coding sequence of functional importance.

**Table 3 | Enrichment of cCREs from ENCODE among low DR regions defined at the 1% and 5% percentiles**

| cCREs[a] | Genome (%) | Enrichment (OR (95% CI)) | |
|---|---|---|---|
| | | DR 1% percentile | DR 5% percentile |
| pELS, CTCF-bound | 0.53 | 6.35 (6.04–6.68) | 3.49 (3.37–3.61) |
| PLS, CTCF-bound | 0.15 | 6.37 (6–6.75) | 3.34 (3.19–3.49) |
| PLS | 0.05 | 2.77 (2.53–3.03) | 1.9 (1.79–2.03) |
| pELS | 0.53 | 2.49 (2.39–2.63) | 1.96 (1.9–2.02) |
| DNase H3K4me3, CTCF-bound | 0.07 | 1.92 (1.67–2.19) | 1.48 (1.38–1.59) |
| dELS, CTCF-bound | 1.86 | 1.65 (1.58–1.71) | 1.53 (1.5–1.57) |
| dELS | 4.11 | 1.17 (1.13–1.2) | 1.27 (1.25–1.3) |
| DNase H3K4me3 | 0.15 | 1.15 (1.04–1.27) | 1.03 (0.974–1.08) |
| CTCF only | 0.47 | 0.878 (0.83–0.925) | 0.96 (0.933–0.987) |

The percentage of the genome covered by candidate *cis*-regulatory elements (cCREs) are indicated for each type of cCRE.

CTCF, CCCTC-binding factor; dELS, distal enhancer-like sequence; OR, odds ratio; pELS, proximal enhancer-like sequence; PLS, promoter-like sequence.

[a]Exons of protein-coding genes found in overlap with cCRE regions were removed.

## Multiple cohorts within UKB

Many GWAS[36] using the UKB data have been based on a subset[5] of 409,559 participants who self-identified as 'white British'. To better leverage the value of a wider range of of UKB participants, we defined three cohorts encompassing 450,690 individuals (Supplementary Table 10), based on genetic clustering of microarray genotypes informed by self-described ethnicity and supervised ancestry inference (Methods). The largest cohort, XBI (Extended Data Fig. 3), contains 431,805 individuals, including 99.6% of the 409,559 prescribed white British set, along with around 23,900 additional individuals previously excluded because they did not identify as white British (thereof 13,000 who identified as 'white Irish'). We believe that this expanded set will increase power in association studies, but have not examined in detail whether this set has other potential benefits or disadvantages. Principal components analysis of the 132,000 XBI individuals with WGS data (Methods), based on 4.6 million loci, reveals an extraordinarily fine-scaled differentiation by geography in the British Irish Isles gene pool (Extended Data Fig. 2).

We defined two other cohorts based on ancestry: African (XAF; $n$ = 9,633; Extended Data Fig. 4) and South Asian (XSA; $n$ = 9,252; Extended Data Fig. 5) (Fig. 3a–c). The 37,598 UKB individuals who do not belong to XBI, XAF or XSA were assigned to the cohort OTH (others). The WGS data of the XAF cohort represent one of the most comprehensive surveys of African sequence variation to date, with reported birthplaces of its members covering 31 of the 44 countries on mainland of sub-Saharan Africa (Extended Data Fig. 4). Owing to the considerable genetic diversity of African populations, and resultant differences in patterns of linkage disequilibrium, the XAF cohort may prove valuable for fine-mapping association signals due to multiple strongly correlated variants identified in XBI or other non-African populations.

We crossed GraphTyperHQ variants with exon annotations and found that, on average, around 1 in 30 individuals is homozygous for rare (minor allele frequency of less than 1%) LoF mutations in the homozygous state and the median number of heterozygous rare LoF is 24 per individual. We detected rare LoF variants in 19,105 genes, in which 2,017 genes had homozygous carriers of rare LoFs (individuals $n$ = 5,102). A marked difference in the number of homozygous LoFs carriers was found between the cohorts, with XSA having the largest fraction of homozygous LoF carriers (Extended Data Fig. 6b). A notable feature of the XSA cohort is elevated genomic inbreeding, probably owing to endogamy[37], particularly among self-identified Pakistani individuals[38] (Extended Data Fig. 6a).

On average, individuals carried alternative alleles of 3,410,510 SNPs and indels (Fig. 4a), per haploid genome. A greater number of variants are generally found in individuals born outside Europe (Extended Data Fig. 7), because the human reference genome is primarily derived from individuals of European ancestry[13]. XAF individuals carry the greatest number of alternative alleles (Fig. 4a). We constructed cohort-specific DRs and found that XAF shows greater depletion around exons than XBI and XSA (Extended Data Fig. 8). Largely owing to variation in the number of individuals sampled, the average number of singletons per individual varies considerably by ancestry (Fig. 4a). Thus, individuals from the XBI, XAF and XSA cohorts have an average of 1,330, 9,623 and 8,340 singleton variants, respectively. In XBI, singleton counts (Fig. 3d) indicate that the expected number of new variants discovered per genome is still substantial, but varies geographically, averaging around 1,000 in northern England and 2,000 in southeastern England. This pattern is largely explained by denser sampling of some regions (Fig. 3e,f) rather than regional ancestry differences.

## Imputation

We were able to reliably impute variants into the entire UKB sample set down to very low frequency (Fig. 4b). We imputed phased genotypes, which permit analysis that depend on phase such as identification of compound LoF heterozygotes. A single reference panel was used to impute into the genomes of all participants in UKB, but results are presented separately for the three cohorts (Supplementary Table 11). This reference panel can be used for accurate imputation in individuals from the UK and many other populations. In the XBI cohort, 98.5% of variants with a frequency above 0.1% and 65.8% of variants in the frequency category of 0.001–0.002% (representing 3–5 WGS carriers) could be reliably imputed (Fig. 4b and Supplementary Fig. 13). Variants were also imputed with high accuracy in XAF and XSA cohorts (Fig. 4b), in which 97.5% and 94.9% of variants in frequencies 1–5% and 56.6% and 48.9% of variants carried by 3–5 sequenced individuals could be imputed, respectively. A larger number of variants, particularly rare ones, are imputed for all cohorts than when using a alternate imputation panel[5] (Supplementary Table 12). It is thus likely that the UKB reference panel provides one of the best-available option for imputing genotypes into population samples from Africa and South Asia.

We found a number of clinically important variants that can now be imputed from the dataset. These include rs63750205 (NM_000518.5 (HBB):c.*110_*111del) in the 3′ UTR of *HBB*, a variant that has been annotated in ClinVar[39] as likely pathogenic for β-thalassaemia. rs63750205-TTA has 0.005% frequency in the imputed XBI cohort (imputation information of 0.98) and is associated with lower mean corpuscular volume by 2.88 s.d. (95% CI 2.43–3.33, two-sided $P = 1.5 \times 10^{-36}$, $\chi^2$).

In the XSA cohort, we found rs563555492-G, a previously reported[40] missense variant in *PIEZO1* (frequency = 3.65% for XSA, 0.046% for XAF and 0.0022% for XBI) associated with higher haemoglobin concentration, effect 0.36 s.d. (95% CI 0.28–0.44, two-sided $P = 8.9 \times 10^{-19}$, $\chi^2$). The variant can be imputed into the XSA population with imputation information of 0.99.

In the XAF cohort, we found the stop-gain variant rs28362286-C (p.Cys679Ter) in *PCSK9* (frequency = 0.93% in XAF, 0.00016% in XBI and 0.0070% in XSA) imputed in the XAF cohort with imputation information of 0.93. The variant lowers non-HDL cholesterol by 0.92 s.d. (95% CI 0.75–1.09, two-sided $P = 2.3 \times 10^{-26}$, $\chi^2$). We found a single homozygous carrier of this variant, who has a 2.5 s.d. lower non-HDL cholesterol than the population mean, is 61 years of age and appears to be healthy.

## SNP and indel associations not in WES

We tested imputed GraphTyper SNP/indel, microsatellite and SV datasets for association with a total of 8,180, 1,291 and 459 phenotypes in the XBI, XAF and XSA cohorts, respectively. We highlight examples of

associations with traits that could not be easily identified in WES or SNP array data, starting with three examples of SNP and indel associations in the XBI cohort.

The first is an association in the XBI cohort between a rare variant—rs117919628-A (frequency = 0.32%; imputation information of 0.90), in the promoter region of *GHRH*, which encodes growth hormone-releasing hormone, close to one of its transcription start sites—and less height (effect = −0.32 s.d. (95% CI 0.27–0.36), two-sided $P = 1.6 \times 10^{-39}$, $\chi^2$). GHRH is a neuropeptide secreted by the hypothalamus to stimulate the synthesis of growth hormone (GH). We note that the effect (−0.32 s.d. or −3 cm) of rs117919628 is greater than any variant reported in large height genome-wide association studies (GWAS; approximately 1,200 associated variants)[41–43]. In addition to reducing height, rs117919628-A is associated with lower serum levels of insulin growth factor 1 (IGF1; effect = −0.36 s.d. (95% CI 0.32–0.40), two-sided $P = 3.2 \times 10^{-58}$, $\chi^2$). The production of IGF1 is stimulated by GH and mediates the effect of GH on childhood growth, further supporting the hypothesis that *GHRH* mediates the effects of rs117919628-A. Owing to its location around 50 bp upstream of the *GHRH* 5′ UTR, this variant is not targeted by the UKB WES, and neither is the only strongly correlated variant rs372043631 (intronic). rs117919628-A is not correlated with rs763014119-C (no individuals carry the minor allele of both variants), a previously reported[44] very rare frameshift deletion in *GHRH* (Phe7Leufster2; frequency = 0.0092%), associated with reduced height and IGF1 levels (height effect = −0.63 s.d (95% CI 0.36–0.89), two-sided $P = 4.6 \times 10^{-6}$; IGF1 effect = −0.74 s.d. (95% CI 0.49–0.99), two-sided $P = 4.9 \times 10^{-9}$, $\chi^2$).

The second example is rs939016030-A, a rare 3′ UTR essential splice acceptor variant in the gene encoding tachykinin 3 (*TAC3*; frequency = 0.033%; c.*2-1G>T in NM_001178054.1 and NM_013251.3). This variant is not found in WES of the UKB[45] and neither are the two highly correlated variants: one intronic (rs34711498) and one intergenic (rs368268673). The minor allele of this 3′ UTR essential splice variant, rs939016030-A, is associated with later age of menarche, with an effect of 0.57 s.d. (95% CI 0.41–0.74) or 11 months (two-sided $P = 1.0 \times 10^{-11}$, $\chi^2$). Rare coding variants in *TAC3* and its receptor *TACR3* have been reported to cause hypogonadotropic hypogonadism[46] under autosomal recessive inheritance. However, in the UKB, the association of the 3′ UTR splice acceptor variant is only driven by heterozygotes (approximately 1 in 1,500 individuals) with no homozygotes detected. We replicated this finding in a set of 39,360 Danish individuals, with an effect of 0.70 s.d. (95% CI 0.34–1.06, frequency = 0.05%, two-sided $P = 0.00014$, $\chi^2$).

The third example is a rare variant (rs1383914144-A; frequency = 0.40%) near the centromere of chromosome 1 (start of 1q) that associates with lower levels of uric acid (effect = −0.43 s.d. (95% CI 0.40–0.46) or −0.58 mg dl$^{-1}$ (95% CI 0.54–0.62), two-sided $P = 8.1 \times 10^{-170}$, $\chi^2$) and protection against gout (OR = 0.36 (95% CI 0.28–0.46), two-sided $P = 4.2 \times 10^{-15}$, $\chi^2$). A second variant, rs1189542743, 4 Mb downstream at the end of chromosome 1p is strongly correlated with rs1383914144 ($r^2 = 0.68$) and yields a similar association with uric acid. No association was reported in this region in the uric acid GWAS[47]. The effect of rs1383914144-A on uric acid is larger than of any variant reported in the latest GWAS meta-analysis of this trait. We replicated these findings in Icelandic individuals (rs1383914144-A, frequency = 0.47%; uric acid: two-sided $P = 8.0 \times 10^{-37}$, $\chi^2$, effect = −0.51 s.d. (95% CI 0.43–0.59); gout: two-sided $P = 0.0018$, $\chi^2$, OR = 0.31 (95% CI 0.15–0.64)).

## Structural variants

We identified SVs in each individual using Manta[48] and combined these with variants from a long-read study[49] and the assemblies of seven individuals[50]. We genotyped the resulting 895,055 SVs (Fig. 4c) with GraphTyper[50], of which 637,321 were considered reliable.

On average, we identified 7,963 reliable SVs per individual, 4,185 deletions and 3,778 insertion (Fig. 4a). These numbers are comparable to

the 7,439 SVs per individual found by Gnomad-SV[51], another short-read study, but considerably smaller than the 22,636 high-quality SVs found in a long-read sequencing study[49], mostly owing to an underrepresentation of insertions and SVs in repetitive regions. SVs show a similar frequency distribution as SNPs and indels and a similar distribution of variants across cohorts (Fig. 4a).

We present four examples of phenotype associations with SVs, not easily found in WES data. First, a rare (frequency = 0.037%) 14,154-bp deletion that removes the first exon in *PCSK9*, previously discovered using long-read sequencing in the Icelandic population and is associated with lower levels of non-HDL cholesterol[49]. There were 32 WGS carriers in the XBI cohort (frequency = 0.012%) and 72 carriers in the XBI imputed set (frequency = 0.0087%) who had 1.22 s.d. (95% CI 0.90–1.55) lower levels of non-HDL cholesterol than non-carriers (two-sided $P = 1.2 \times 10^{-13}$, $\chi^2$).

The second example is a 4,160-bp deletion (frequency = 0.037% in XBI) that removes the promoter region from 4,300 to 140 bp upstream of the *ALB* gene, which encodes albumin. Not surprisingly, carriers of this deletion have markedly lower levels of serum albumin (effect = 1.50 s.d. (95% CI 1.35–1.62), two-sided $P = 9.5 \times 10^{-118}$, $\chi^2$). The variant is also associated with traits correlated with albumin levels; carriers had lower levels of calcium and cholesterol: 0.62 s.d. (95% CI 0.50–0.75), two-sided $P = 2.9 \times 10^{-22}$, $\chi^2$) and 0.45 s.d. (95% CI 0.30–0.59, two-sided $P = 1.1 \times 10^{-9}$, $\chi^2$), respectively.

The third SV example is a 16,411-bp deletion (frequency = 0.0090% in XBI) that removes the last two exons (4 and 5) of *GCSH*, which encodes glycine cleavage system H protein. Carriers of this deletion have markedly higher levels of glycine in the UKB metabolomics dataset (effect = 1.45 s.d. (95% CI 1.01–1.86), two-sided $P = 1.2 \times 10^{-10}$, $\chi^2$).

The final example is a rare (frequency = 0.892% in XBI) 754-bp deletion overlapping exon 6 of *NMRK2*, which encodes nicotinamide riboside kinase 2, that removes 72 bp from the transcribed RNA that corresponds to a 24 amino acid in-frame deletion in the translated protein. Carriers of this deletion have a 0.22 s.d. (95% CI 0.18–0.27) earlier age at menopause (two-sided $P = 1.1 \times 10^{-26}$, $\chi^2$). Nearby is the variant rs147068659, which has been reported to be associated with this trait[52], with an effect of 0.20 s.d. (95% CI 0.16–0.24) earlier age at menopause (two-sided $P = 2.0 \times 10^{-20}$, $\chi^2$) in the XBI cohort. The deletion and rs147068659 are correlated ($r^2 = 0.67$); after conditional analysis the deletion remains significant (two-sided $P = 6.4 \times 10^{-8}$, $\chi^2$), whereas rs147068659 does not (two-sided $P = 0.39$, $\chi^2$), indicating that the deletion is the lead variant for the locus. *NMRK2* is primarily expressed in heart and muscle tissue[53]. In our dataset of right atrium heart tissue, one individual out of a set of 169 RNA-sequenced individuals is a carrier of this deletion. As expected, we observed decreased expression of exon 6 in this individual and an increase in the fraction of transcript fragments skipping exon 6 (Extended Data Fig. 9).

## Microsatellites are commonly overlooked

We identified 14,321,152 alleles at 2,536,688 microsatellite loci using popSTR[54] in the 150,119 WGS individuals who carry, on average, 810,606 non-reference microsatellite alleles. The number of non-reference alleles carried per individual shows a similar distribution across the UKB cohorts as other variant types characterized in this study (Fig. 4a). Microsatellites are among the most rapidly mutating variants in the human genome and a source of genetic variation that is usually overlooked in GWAS. Repeat expansions are known to associate with a number of phenotypes, including fragile X syndrome[55]. We were able to impute microsatellites down to a very low frequency (Supplementary Fig. 4) in all three cohorts, providing one of the first large-scale datasets of imputed microsatellites.

We genotyped a microsatellite within the *CACNA1A* gene, which encodes voltage-gated calcium channel subunit-α 1A. Individuals who have 20 or more repeats of this microsatellite generally suffer from

lifelong conditions that affect the brain, including familial hemiplegic migraine type 1, epilepsy, episodic ataxia type 2 and spinocerebellar ataxia type 6 (ref. [56]). Carriers in the XBI cohort of 22 copies of the microsatellite repeat were at greater risk for hereditary ataxia (frequency = 0.0071%, OR = 304, two-sided $P = 1.1 \times 10^{-31}$, $\chi^2$).

We also confirmed an association between a microsatellite within the 3′ UTR of *DMPK*, which encodes DM1 protein kinase, and myotonic dystrophy in the XBI cohort. Expression of *DMPK* has been shown to be negatively correlated with the number of repeats of the microsatellite[57]. The risk of myotonic dystrophy increases with copy number of the repeats, rising rapidly with the number of repeats carried by an individual up to an OR of 161 for individuals carrying 39 or more repeats (Extended Data Fig. 10 and Supplementary Table 13).

## Discussion

The dataset provided by sequencing the whole genomes of approximately 150,000 UKB participants is unparalleled in its size and provides the most extensive characterization of the sequence diversity in the germline genomes of a single population to date. We characterized an extensive set of sequence variants in the WGS individuals, providing two sets of SNP and indel data, as well as microsatellite and SV data, variant classes that are frequently not interrogated in GWAS. The number of SNPs and indels are 40-fold greater than from WES of the same individuals. Even within annotated coding exons, WES misses 10.7% of variants, found through WGS. WES misses most of the remainder of the genome, including functionally important UTRs, promoter regions and exons yet to be annotated. The importance of these regions is exemplified by the discovery of rare non-coding sequence variants with larger effects on height and menarche than any variants described in GWAS to date.

We expect the DR score presented here to be an important resource for identifying genomic regions of functional importance, although further evaluations should be taken to understand its properties and implications and how it compares to other measures of conservation and sequence constraint. Although coding exons are clearly under strong purifying selection, as represented by a low DR score, they represent only a small fraction of the regions with a low DR score. The large-scale sequencing described here, as well as the continued effort in sequencing the entire UKB, promises to vastly increase our understanding of the function and impact of the non-coding genome. When combined with the extensive characterization of phenotypic diversity in the UKB, these data should greatly improve our understanding of the relationship between human genome variation and phenotype diversity.

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

**DBDS Genetic Consortium**

**Steffen Andersen[13], Karina Banasik[9], Søren Brunak[9], Kristoffer Burgdorf[10], Maria Didriksen[10], Khoa Manh Dinh[8], Christian Erikstrup[8], Daniel Gudbjartsson[1], Thomas Folkmann Hansen[14], Henrik Hjalgrim[15], Gregor Jemec[16], Poul Jennum[17], Pär Ingemar Johansson[10], Margit Anita Hørup Larsen[10], Susan Mikkelsen[8], Kasper Rene Nielsen[18], Mette Nyegaard[19], Sisse Rye Ostrowski[10], Ole Birger Pedersen[6], Kari Stefansson[1], Hreinn Stefansson[1], Susanne Sækmose[6], Erik Sørensen[10], Unnur Thorsteinsdottir[1], Mie Topholm Brun[20], Henrik Ullum[21] & Thomas Werge[22]**

[13]Department of Finance, Copenhagen Business School, Copenhagen, Denmark. [14]Danish Headache Center, Department of Neurology, Copenhagen University Hospital, Rigshospitalet–Glostrup, Glostrup, Denmark. [15]Department of Epidemiology Research, Statens Serum Institut and Centre for Cancer Research, Danish Cancer Society, Copenhagen, Denmark. [16]Department of Dermatology, Zealand University Hospital-Roskilde, Roskilde, Denmark. [17]Department of Clinical Neurophysiology, University of Copenhagen, Copenhagen, Denmark. [18]Department of Clinical Immunology, Aalborg University Hospital, Aalborg, Denmark. [19]Department of Biomedicine, Aarhus University, Aarhus, Denmark. [20]Department of Clinical Immunology, Odense University Hospital, Odense, Denmark. [21]Statens Serum Institute, Copenhagen, Denmark. [22]Institute of Biological Psychiatry Mental Health Centre, Sct. Hans, Copenhagen University Hospital-Roskilde, Roskilde, Denmark.

## Methods

### Datasets

**UKB data.** The UKB phenotype and genotype data were collected following informed consent obtained from all participants. The North West Research Ethics Committee reviewed and approved the scientific protocol and operational procedures (REC reference number: 06/MRE08/65) of the UKB. Data for this study were obtained and research conducted under the UKB applications license numbers 24898, 52293, 68574 and 69804. Sequence data were processed as described in Supplementary Notes 1–4, Supplementary Figs. 5–8 and Supplementary Tables 16 and 17.

Phenotypes were downloaded from the UKB. A total of 8,180, 1,291 and 459 phenotypes were constructed for the XBI, XAF and XSA cohorts, respectively. The examples presented here were selected as noteworthy representative examples of association. The processing of phenotypes presented here, with reference to the field identity in the UKB data showcase, is provided in Supplementary Table 15.

**Icelandic data.** The gout sample set[60], a total of 1,740 Icelandic individuals, was recruited through multiple sources. A subset of these individuals were regular users of anti-gout medication corresponding to the Anatomical Therapeutic Chemical Classification System class M04 (ATC-M04). Individuals using ATC-M04 were identified through questionnaires at the time of entry into genetics projects at deCODE and provided by the Directorate of Heahth from entry in the Prescription Medicines Register (2005-2020) or the Register of RAI Assessments and Minimum Data Set (MDS) for residents and applicants of nursing homes (1993–2018). Furthermore, about one-half had received a clinical diagnosis of gout (International Classification of Disease: ICD-9 code 274 or ICD-10 code M10) between 1984 and 2019 at Landspitali, the National University Hospital of Iceland, or at two rheumatology clinics, or such a diagnosis was determined by examining RAI and MDS medical records.

Serum levels of uric acid in blood samples from 95,086 Icelandic individuals were obtained from Landspitali, the National University Hospital of Iceland, and the Icelandic Medical Center (Laeknasetrid) Laboratory in Mjodd (RAM) between 1990 and 2020. Serum levels of uric acid were normalized to a standard normal distribution using quantile–quantile normalization and then adjusted for sex, year of birth and age at measurement. For individuals for whom more than one measurement was available, we used the average of the normalized value. Serum levels of uric acid were determined from an enzymatic reaction in which uricase oxidizes urate to allantoin and hydrogen peroxide, which, with the aid of peroxidase and a dye, forms a coloured complex that can be measured in a photometer at a wavelength of 670 nm.

All participating individuals who donated blood signed informed consent. The identities of participants were encrypted using a third-party system approved and monitored by the Icelandic Data Protection Authority. The study was approved by the National Bioethics Committee of Iceland (approval no. VSN-15-023) following evaluation of the Icelandic Data Protection Authority. All data processing complies with the instructions of the Data Protection Authority (PV_2017060950ÞS).

RNA sequence data analysis was approved by the Icelandic Data Protection Authority and the National Bioethics Committee of Iceland (no. VSNb2015030021).

**Danish data.** Data were provided from the Danish Blood Donor Study (DBDS)[61]. The DBDS genetic study has been approved by the Danish National Committee on Health Research Ethics (NVK-1700407) and by the Danish Capital Region Data Protection Office (P-2019-99).

### SNP and indel calling with GraphTyper

Before running GraphTyper, we preprocessed all input compressed reference-oriented alignment map (CRAM) index (CRAI) indices by extracting a large single file containing all CRAI index entries with sample ID for a 50-kb window (with 1-kb padding at each side of the region) for all samples. For each region, we then created a chopped CRAI for each sample by processing the large file for the corresponding region, substantially reducing the amount of CRAI index entries read.

Furthermore, we created a sequence cache of the reference FASTA file using the 'seq_cache_populate.pl' script distributed with samtools 1.9. In each region, we copied the corresponding sequence cache to the local disk and used it for reading the CRAM files by setting the 'REF_CACHE' environment variable.

We ran GraphTyper (v2.7.1) using the 'genotype' subcommand. The full command that we ran was in the format:

graphtyper genotype ${UKBIO_REFERENCE} --sams=${SAMS} --sams_index=${CRAI_TMP}/crai_filelist.txt --avg_cov_by_readlen=${COVERAGES} --region=${REGION} --threads=${THREADS} --verbose

Where UKBIO_REFERENCE is the GRCh38_full_analysis_set_plus_decoy_hla FASTA sequence file, SAMS is a list of all input BAM/CRAM files, CRAI_TMP is a path to the chopped CRAI files on the local disk, COVERAGES is the coverage divided by the read length for each input file, REGION is the genotyping region and THREADS is the number of threads to use.

SNP and indel calling with GATK is given in Supplementary Note 5. Detailed comparisons of GraphTyper and GATK call sets are provided in Supplementary Notes 6 and 7, Supplementary Figs. 9–12 and Supplementary Tables 18–21.

**Running time.** All jobs were run using 12 cores with 60 GB of reserved RAM. Approximately 1% of jobs were rerun using 24 cores with 120 GB reserved RAM. A few jobs requiring more cores and memory, with a single job finishing with 48 cores and 1,000 GB of RAM. Total reserved CPU time on cluster was 5.8 million CPU hours and total effective compute time of 5.0 million CPU hours. The difference in these numbers is explained by the fact that not all cores reserved for the program may not utilize all at the same time.

### SV calling with Manta and GraphTyper

We ran a SV genotyping pipeline similar to the one that we had previously applied to 49,962 Icelandic individuals[50]. In summary, we ran Manta[48] v1.6 to discover SVs on all 150,119 individuals in the genotyping set. We also created a set of highly confident common SVs (imputation information above 0.95, with frequency above 0.1%) from our previous studies using both Illumina short reads[50] and Oxford Nanopore long-read data[49]. Finally, we inferred a set of SVs from six publicly available assembly datasets using dipcall[62], as previously described[50]. We used svimmer[50] to merge these different SV datasets and we called the resulting SVs using GraphTyper[50] version 2.7.1. By incorporating data from long-read data and high-quality assemblies, we were able to call more true SVs than using short reads only, particularly for common SVs.

A total of 895,054 variants were called, of which 637,321 variants were annotated as "Pass". Variant counts are presented for variants annotated by GraphTyper as "Pass", unless otherwise noted.

The majority of the SVs are deletions (81.3%); however, we observed only slightly more deletions than insertions and duplications on average per individual (Fig. 4a). This is because the source for many insertions are long reads and assembly data, and thus many rare insertions are missing. Deletions are typically easier to discover in short-read data. Individuals who belong to the XAF cohort carry more SVs than in the other cohorts (Fig. 4b).

### Imputation and phasing

The UKB samples were SNP chip genotyped with a custom-made Affymetrix chip, UK BiLEVE Axiom, in the first 50,000 individuals[63], and the Affymetrix UKB Axiom array[64] in the remaining participants. We used the existing long-range phasing of the SNP chip-genotyped samples[5].

We excluded SNP and indel sequence variants in which at least 50% of the samples had no coverage (GQ score = 0), if the Hardy–Weinberg

*P* value was less than $10^{-30}$ or if heterozygous excess was less than 0.5 or greater than 1.5.

We used the remaining sequence variants and the long-range-phased chip data to create a haplotype reference panel using in-house tools[1,65]. We then imputed the haplotype reference panel variants into the chip-genotyped samples using in-house tools and methods previously described[1,65].

The imputation consists of estimating, for each haplotype, haplotype sharing with haplotypes in the haplotype reference panel, giving haplotype weights for each haplotype. These weights along with allele probabilities for each haplotype in the haplotype reference panel allow imputation with a Li and Stephens[66] model similar to the one used in IMPUTE2 (ref. [67]). Estimation of haplotype weights was based on long-range-phased chip haplotypes.

Sequence variant phasing consists of iteratively imputing the phase in each sequenced sample based on the other sequenced samples and the estimated phase from the last iteration. The imputed genotypes, along with the original genotypes, are weighted together to estimate new allele probabilites for the haplotypes. Imputation is done as described above.

We computed a leave-one-out $r^2$ score (L1oR2) as the squared correlation ($r^2$ value) of the original genotype calls, with the genotypes imputed for each sample when excluding the original genotype of the sample from the imputation input.

Batch effects from the sequencing centre were discovered in both raw genotype (Supplementary Table 21) and imputed data (Supplementary Table 22).

## Identification of functionally important regions

To identify functionally important regions, we started by estimating whether reliable basecalls can be expected to be made at each site in the genome. The sequence coverage at each base pair in GRCh38 was computed for each of the 1,000 randomly selected individuals. At each base pair, we then computed the mean and s.d. of coverage across the 1,000 individuals. Base pairs with mean coverage of at least 20 and s.d. coverage of at most 12 were considered reliable base pairs. Only variants in GraphTyperHQ (AAscore > 0.5) were considered in the analysis.

**Recurrent mutations and spectra under saturation.** Using the classification of SNP variants from above, we calculated the ratio of all SNPs in GraphTyperHQ that falls into each category. Then, we did the same restricting to singletons, that is, calculate the proportion of singletons falling into each mutation class. For comparison, we calculated the fractions of each SNP class in all 181,258 SNPs from a curated list of 194,687 de novo mutations in 2,976 Icelandic trios[20]. We used this distribution on mutation classes to calculate the transition to tranversion ratio in each case.

To get a list of recurrent mutations, we joined this list of de novo mutations with GraphTyperHQ.

**Saturation for general mutation classes.** We restricted our analysis to the reliable base pairs described above and grouped base pairs and their complement and considered each A or T base in the genome as a mutation opportunity for T>A, T>C or T>G mutations. Similarly, we considered each G or C base as a potential C>A, C>G or C>T mutation, splitting C>T into two classes based on whether they occur in a CpG context. We then computed the saturation ratio as the number of observed mutations in GraphTyperHQ divided by the number of mutation opportunities at reliable base pairs. Computation was done separately for the autosomes and chromosome X. 95% CIs were computed using a normal approximation to the binomial distribution, treating each site as an independent observation.

**Sites methylated in the germline.** We determined sites on GRCh38 that are methylated in the germ line using ENCODE whole-genome bisulfite sequencing[9] data from samples of human testes and ovaries. More precisely, we used sample ENCFF946UQB and ENCFF157ZPP for testes and ENCFF561KYJ, ENCFF545XYI and ENCFF515OOQ for ovaries.

We assumed that methylation is strand symmetric and computed the methylation ratio for each CpG dinucleotide in a given tissue type by tabulating the number of reads supporting methylation or non-methylation in each dinucleotide, summing over all samples of a given tissue type and then computed the fraction of reads that support methylation.

We considered a site in a CpG dinucleotide on the reference genome methylated in the germ line if its methylation ratio was at least 0.7 in both testes and ovaries, and the combined depth was at least 20 for testes and 30 for ovaries, or 10 times the number of samples in each tissue type. This resulted in a list of 17,902,255 CpG (17,345,777 autosomal) dinucleotides, with 35,804,510 (34,691,554 autosomal) CpG>TpG mutation opportunities.

**Saturation at methylated CpG sites.** For each potential CpG>TpG at a methylated site, we assessed its most significant potential consequence with Variant Effect Predictor[68] v. 100. In the case of multiple such consequences, we chose the alphabetically last one. We also classified them based on the functional classifications described above. For each class, we estimated the saturation as the ratio of variants of that functional class in GraphTyperHQ divided by the number of mutation opportunities. 95% CIs were computed using a normal approximation to the binomial distribution, treating each site as an independent observation.

**Depletion rank.** We followed a methodology akin to a previously published study[27]. A variant depletion score was computed for an overlapping set of 500-bp windows in the genome with a 50-bp step size. A total of 49,104,026 500-bp windows in which at least 450 bp were considered reliable base pairs were considered for further analysis. We tallied the number of occurrences of each possible heptamer (H) and the number of times the central base pair in the heptamer was observed as a SNP (S), across the first set of non-overlapping windows. To account for regional mutational patterns in the genome[69], we dichotomized the genome into two mutually exclusive subsets, inside and outside C>G-enriched regions (Supplementary Table 12 in ref. [69]). The ratio S:H was then interpreted as the expected mutation rate of the heptamer, separately for each of the two subsets. For each window, we then computed the observed number of variants (O) and then subtracted its expected number of variants (E), given its heptamers. This difference was divided by the square root of the expected value ((O−E)/√E). We exclued windows from the analysis in which the average AAscore was lower than 0.85 for variants within the window. These ((O−E)/√E) numbers were then sorted and the window with the *i*-th lowest depletion score was assigned a DR of $100(i-0.5)/n$, where *n* is the total number of windows.

To compute DR restricted to the cohorts, we applied the same approach restricting to sequence variants that are present in each of the XBI, XSA and XAF cohorts.

## Association testing

We tested for association with quantitative traits based on the linear mixed model implemented in BOLT-LMM[70]. We used BOLT-LMM to calculate leave-one-chromosome out residuals, which we then tested for association using simple linear regression. We used logistic regression to test for the association between sequence variants and binary traits. We tested variants for association under the additive model using the expected allele counts as a covariate for quantitative traits and integrating over the possible genotypes for binary traits. Sequencing status (whether the individual is one of the WGS individuals) and other available individual characteristics that correlated with the trait were also included in the model: sex, age and principal components (20 for XBI and XAF, 45 for XSA) to adjust for population stratification. Association analyses with XAF and XSA ethnicities have sample sizes of

less than 10,000 and therefore were done with linear regression directly instead of BOLT-LMM. The correction factor used was the intercept of each regression analysis.

We used linkage disequilibrium (LD) score regression to account for distribution inflation in the dataset due to cryptic relatedness and population stratification[71]. Using 1.1 million variants, we regressed the $\chi^2$ statistics from our GWAS against the LD score and used the intercepts as a correction factor. Effect sizes based on the leave-one-chromosome out residuals were shrunk and we rescaled them based on the shrinkage of the 1.1 million variants used in the LD score regression. Supplementary Table 24 lists statistics for the GWAS analysis of each of the association signals presented here. Manhattan plots, quantile–quantile plots and histograms of inverse-normal-transformed values after adjustment for covariates age, sex and 40 principal components can be found in Supplementary Figs. 14 and 15 for quantitative and binary phenotypes, respectively. Locus plots for uric acid and menarche association can be found in Supplementary Fig. 16. OMIM[32] and Open Targets[72] annotations of the genes presented are provided in Supplementary Table 14.

No statistical methods were used to predetermine sample size for association testing. All associations reported are for imputed genotypes. For comparison purposes, associations were also performed on the genotypes directly. For the association testing perfomed on the directly genotyped markers, the same set of covariates were used, apart from sequencing status (as all individuals were sequenced), and also the sequencing centre (deCODE, Sanger main, Sanger Vanguard) was used as a covariate. Supplementary Table 25 shows the correlation between the raw and the imputed genotypes and batch effects for sequencing centre in the XBI cohort.

An individual was deemed to be a carrier of an allele if the probability that the individual carried the allele was at least 0.9. The association analysis was limited to markers in which at least one (XAF, XSA), two (XBI, imputed dataset) or three (XBI, raw genotypes) individuals carried the minor allele. As association tests are frequently limited to a subset of the individuals in the dataset, the association analysis was further limited to those markers in which there was at least one carrier among the individuals in the association test. In the imputed dataset, association tests were further limited to those markers with imputation information > 0.5 and in the raw genotype set to those markers with sequencing information > 0.8 (ref. [1]).

### Defining cohorts

Most studies of UKB data to date have been conducted on a list of 409,554 'white British' individuals created by the UKB on the basis of white British self-identification and clustering on genetic principal components derived from microarray genotypes[5]. Like some recent studies[44,73,74], we wished to capitalize on the diversity in the UKB. To achieve this, we defined three cohorts based on the most common ancestries identified among the participants, using a combination of (1) uniform manifold approximation and projection (UMAP) dimension reduction of 40 genetic principal components provided by UKB, and (2) ADMIXTURE analysis supervised on five reference populations and self-reported ethnicity information.

To define the three cohorts, we followed previous work[75] and applied UMAP to the 40 genetic principal components provided by the UKB. UMAP was performed in R using umap::umap() using default parameters in v0.2.3, notably, n_neighbours 15 and min_dist 0.1. UMAP placed the individuals in a two-dimensional latent space featuring several clusters and filaments. These structures showed a correspondence with self-described ethnicity (Supplementary Fig. 17).

To provide a separate measure of ancestry that we could use to inform our interpretation of the UMAP clusters, we superimposed results from a supervised ADMIXTURE[58] analysis of the UKB microarray genotypes (Supplementary Section ADMIXTURE), using five training populations from the 1000 Genomes Project[8]: CEU (northern Europeans from Utah), CHB (Han Chinese in Beijing), ITU (Indian Telugu in

the UK), PEL (Peruvians in Lima) and YRI (Yoruba in Ibadan, Nigeria). We observed a clear correspondence between UMAP coordinates and ancestry proportions assigned by ADMIXTURE (Supplementary Figs. 18 and 19). Using this correspondence and guided by self-reported ethnicity information, we defined the cohorts by manually delineating regions in the UMAP latent space that were limited to individuals with British–Irish ancestry (XBI; $n$ = 431,805), South Asian ancestry (XSA; $n$ = 9,633) and African ancestry (XAF; $n$ = 9,252). This left 37,598 individuals with genotype data, who were assigned to an arbitrary cohort that we refer to as OTH (for other). The distribution of ancestry was estimated using ADMIXTURE in each of the four cohorts (Supplementary Fig. 18).

The most systematic difference between the XBI cohort and the prevailing UKB-defined white British set is our inclusion in XBI of around 12,500 individuals identifying as white Irish. This is clearly justified, given the known geographical and cultural proximity of the populations of Britain and the island of Ireland. More importantly, both our analyses (and those of previous publications) clearly reveal evidence for extensive gene flow between them. Thus, the main Irish genetic cluster appears in principal components analysis as an integrated component of continuous variation in the UK (Extended Data Fig. 2), and is not clearly separated from others. Another major difference of the XBI cohort relative to the much-used white British set, is the addition of around 10,900 individuals who did not identify as white British, but we infered to have ancestry indistinguishable from British–Irish individuals. We note that the greater size of the XBI cohort should provide more statistical power to detect genotype–phenotype associations. Cohort definitions are described in further detail in Supplementary Notes 16–22 and Supplementary Figs. 20–22.

### Reporting summary

Further information on research design is available in the Nature Research Reporting Summary linked to this paper.

### Data availability

WGS, genotype data, phased and imputed data can be accessed via the UKB research analysis platform (RAP): https://ukbiobank.dnanexus.com/landing. The Research Analysis Platform is open to researchers who are listed as collaborators on UKB-approved access applications. Summary statistics for GWAS can be downloaded, for scientific purpose only, at https://www.decode.com/summarydata/. The DR score is included as supplementary data. Summary statistics for the Danish replication phenotype can be made available on request to O.B.P. Summary statistics for the Icelandic replication phenotype can be made avaliable on request to K.S. The human reference genome GRCh38 can be found at: http://ftp.1000genomes.ebi.ac.uk/vol1/ftp/technical/reference/GRCh38_reference_genome/. Genome in a Bottle WGS samples can be found at: https://ftp-trace.ncbi.nlm.nih.gov/ReferenceSamples/giab/data/. ENSEMBL: https://m.ensembl.org/info/data/mysql.html.

### Code availability

We used publicly available software (URLs are listed below) in conjunction with the above described algorithms. BamQC (v 1.0.0): https://github.com/DecodeGenetics/BamQC. GraphTyper (v2.7.1): https://github.com/DecodeGenetics/graphtyper. GATK resource bundle (v4.0.12): gs://genomics-public-data/resources/broad/hg38/v0. Svimmer (v0.1): https://github.com/DecodeGenetics/svimmer. popSTR (v2.0): https://github.com/DecodeGenetics/popSTR. Dipcall (v0.1): https://github.com/lh3/dipcall. RTG Tools (v3.8.4): https://github.com/RealTimeGenomics/rtg-tools. bcl2fastq (v2.20.0.422): https://support.illumina.com/sequencing/sequencing_software/bcl2fastq-conversion-software.html. Samtools (v1.9): http://www.

htslib.org/. Samblaster (v0.1.24): https://github.com/GregoryFaust/samblaster. We used R (v3.6.0; https://www.r-project.org/) extensively to analyse data and create plots.

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

**Acknowledgements** We thank the participants of the UKB. The sequencing of 450,000 WGS individuals from the UKB, including the 150,119 described here has been funded by the UKB WGS consortium consisting of the UK Government's research and innovation agency, UK Research and Innovation (UKRI), through the Industrial Strategy Challenge Fund, The Wellcome Trust and the pharmaceutical companies Amgen, AstraZeneca, GlaxoSmithKline and Johnson & Johnson. DNA sequencing was performed at the Welcome Trust Sanger Institute and deCODE genetics.

**Author contributions** The paper was written by B.V.H. and K.S., with input from H.P.E., K.H.S.M., O.E., D.F.G., O.T.M., G.M., U.T., A.H., H.J. and P.S. K.H.S.M. and A.H. defined the cohorts. O.E. and H.J. identified functionally important regions. F.J. and U.T. were responsible for laboratory operations. DNA sequencing was performed by D.N.M., S.S., K.K. and O.T.M. Sample isolation was performed by E.S. and J.S. G.M. was responsible for the sequence analysis pipeline, developed by B.V.H., A.G., P.I.O., M.A., S.T.S., F.Z. and S.A.G., and run by G.T.S. B.V.H., H.P.E., H.Hauswedell, G.P., S.K., G.H. and S.A.G. developed analysis tools. Association analysis was performed by B.V.H., M.O.U., A.O., B.O.J., S.K., B.D.S., D.B., V.T., U.S. and P.S. Phenotypes were defined by M.O.U., V.T., G.T., I.J., T.R., H.Holm, H.S. and P.S. SNP and SV genotyping was performed by H.P.E., P.I.O. and B.S. Microsatellite genotyping was performed by S.K. Data analysis was performed by B.V.H., H.P.E., H.Hauswedell, G.P., A.O., O.A.S., G.S. and K.N. RNA sequencing data were analysed by G.H.H., supervised by P.M. D.F.G. supervised association data and DR analysis. Figures were drawn by M.T.H. and K.H.S.M. H.J., A.J.G., I.O. and P.J. collected clinical data in Iceland. O.B.P., C.E., S.B., S.R.O. and the DBDSGC collected clinical data in Denmark. The study was supervised by B.V.H. and K.S. All authors agreed to the final version of the manuscript.

**Competing interests** B.V.H., H.P.E., K.H.S.M., H.Hauswedell, O.E., M.O.U., G.P., M.T.H., A.O., B.O.J., S.K., B.D.S., O.A.S., D.B., G.H., V.T., A.G., P.I.O., F.Z., M.A., S.T.S., B.S., S.A.G., G.T.S., G.H.H., G.S., K.N., U.S., D.N.M., S.S., K.K., E.S., G.T., F.J., P.M., I.J., T.R., H.Holm, H.S., J.S., D.F.G., O.T.M., G.M., U.T., A.H., H.J., P.S. and K.S. are employees of deCODE genetics/Amgen.

**Additional information**
**Correspondence and requests for materials** should be addressed to Bjarni V. Halldorsson or Kari Stefansson.

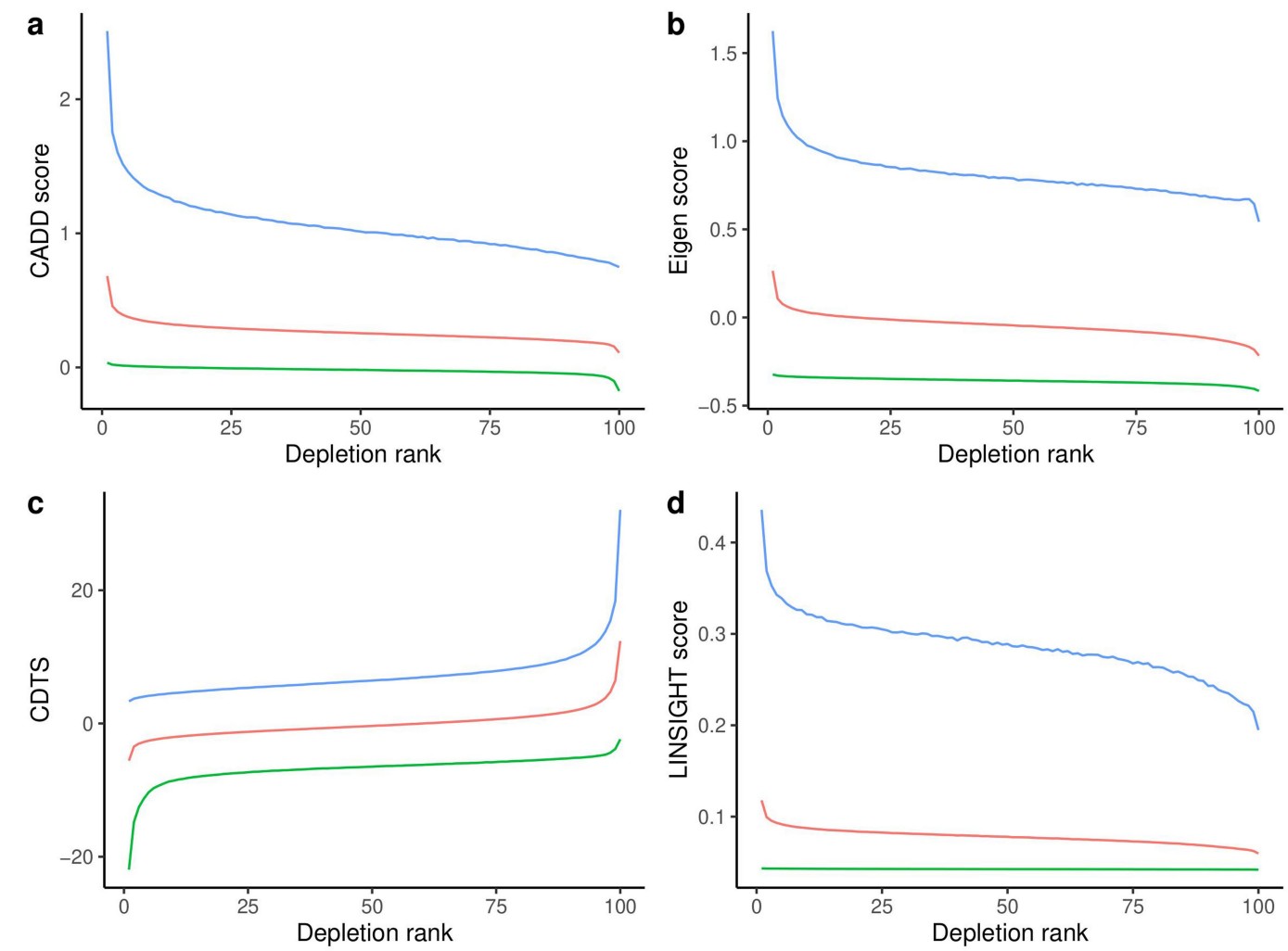

**Extended Data Fig. 1 | Average score in 500bp windows as a function of Depletion Rank for. a**, CADD, **b**, Eigen, **c**, CDTS, and **d**, LINSIGHT. Green line represents average score, blue and red line 95-th percentile.

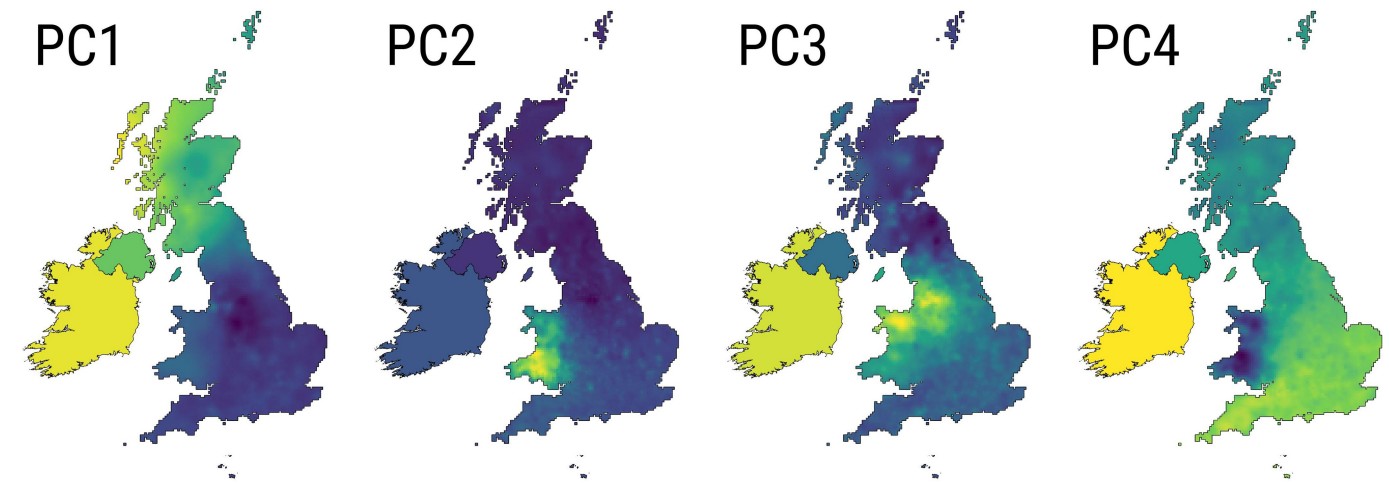

**Extended Data Fig. 2** | Geographic distribution of the loadings of the first four principal components of a PCA of the XBI population.

# XBI

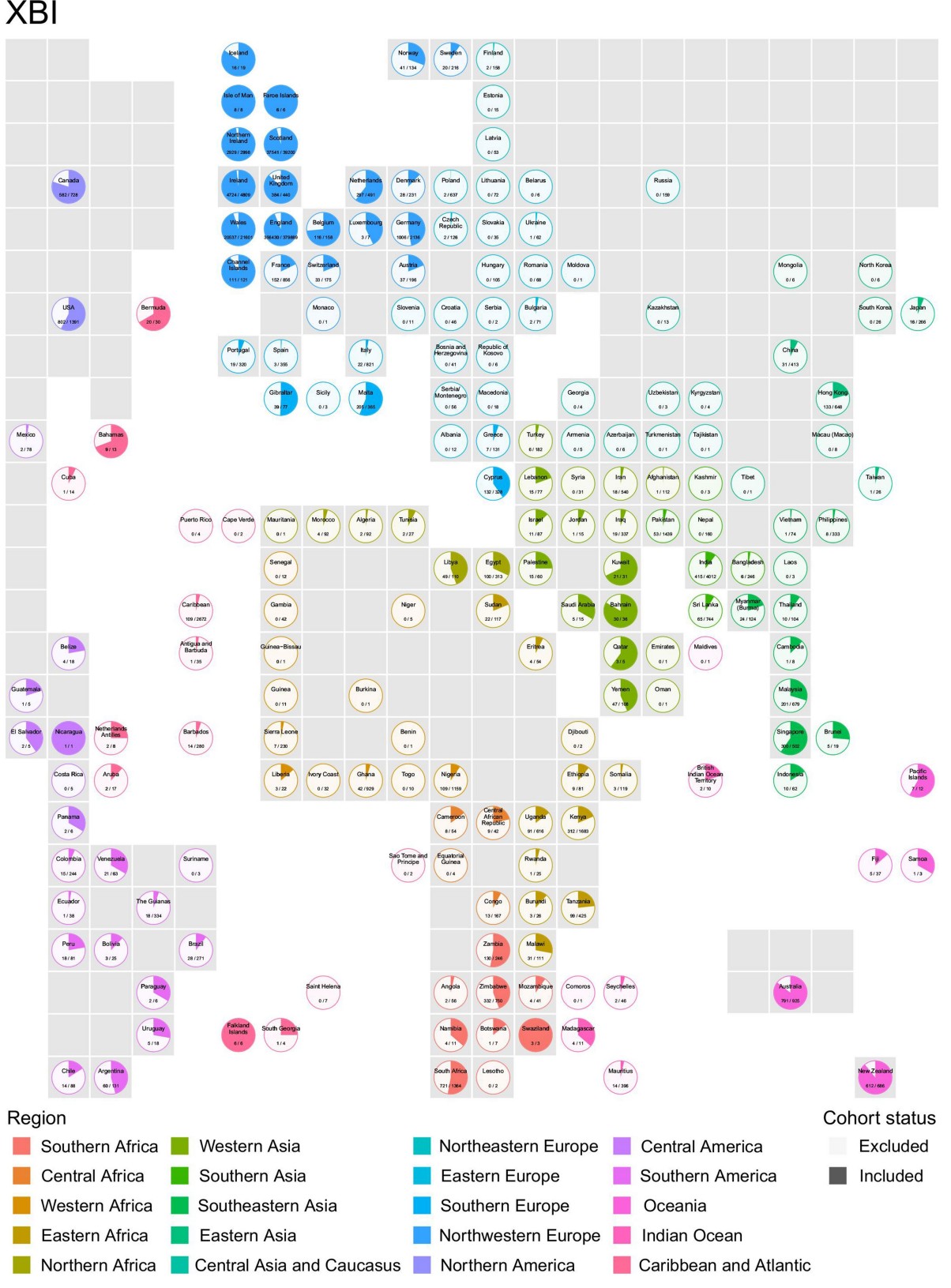

**Extended Data Fig. 3 | Cartogram-pies indicating the proportion of individuals born in each country (name shown on top of pies) in the XBI cohort. Pies are** placed roughly according to their country's position on a world map. Grey and white squares represent sea and land respectively.

## XAF

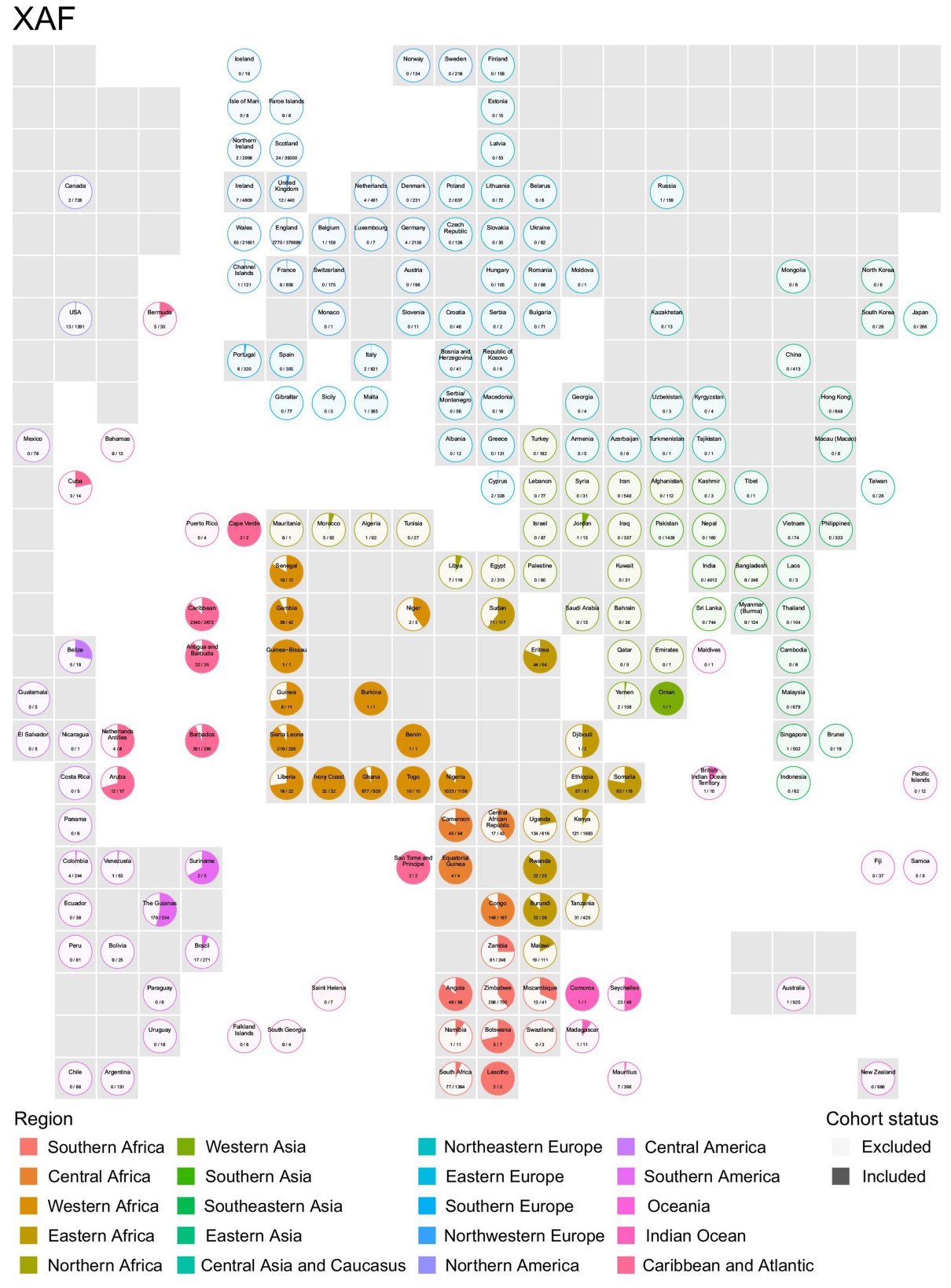

**Extended Data Fig. 4 | Cartogram-pies indicating the proportion of individuals born in each country (name shown on top of pies) in the XAF cohort.** Pies are placed roughly according to their country's position on a world map. Grey and white squares represent sea and land respectively.

# XSA

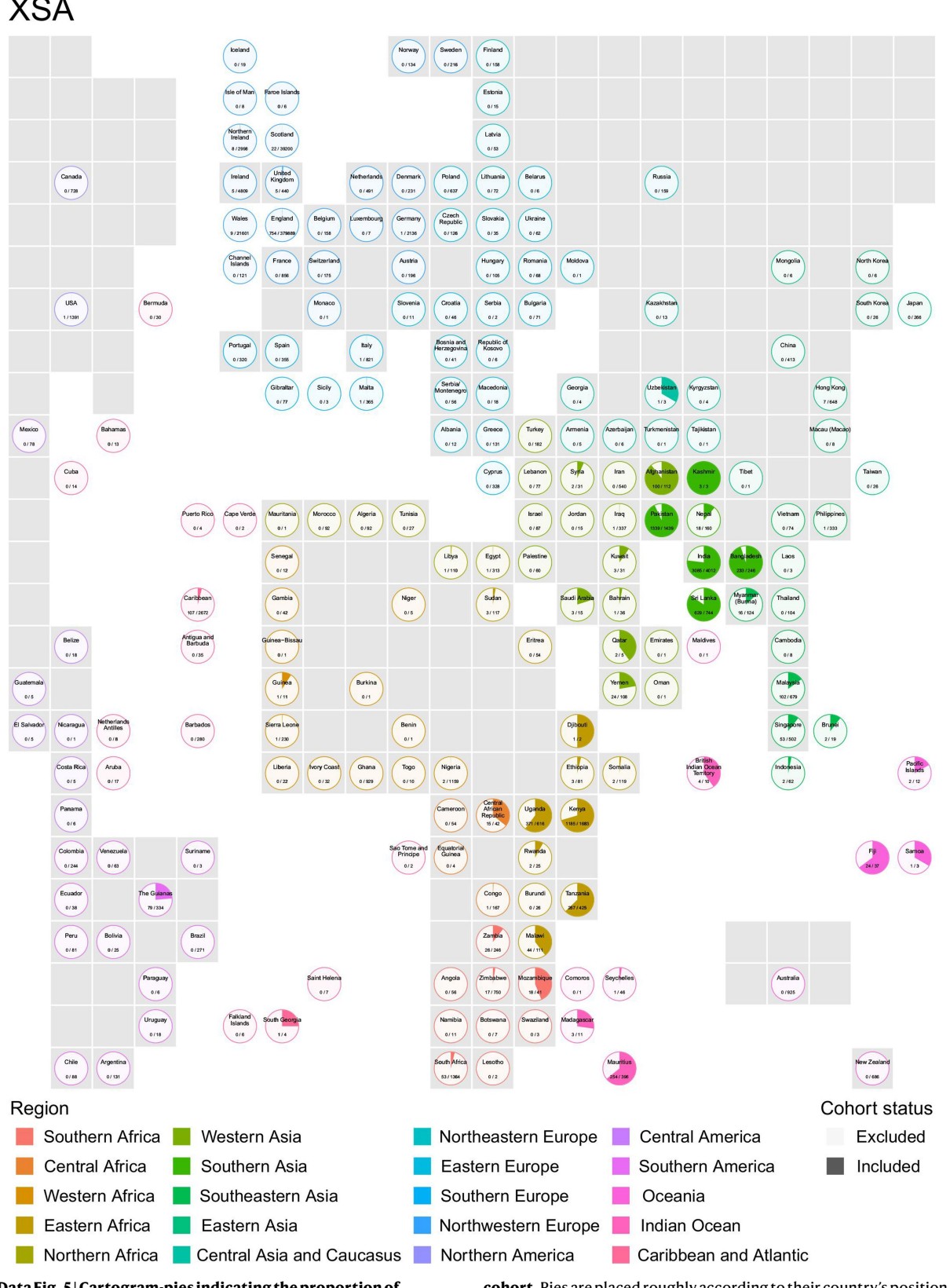

**Region**

| | | | |
|---|---|---|---|
| ■ Southern Africa | ■ Western Asia | ■ Northeastern Europe | ■ Central America |
| ■ Central Africa | ■ Southern Asia | ■ Eastern Europe | ■ Southern America |
| ■ Western Africa | ■ Southeastern Asia | ■ Southern Europe | ■ Oceania |
| ■ Eastern Africa | ■ Eastern Asia | ■ Northwestern Europe | ■ Indian Ocean |
| ■ Northern Africa | ■ Central Asia and Caucasus | ■ Northern America | ■ Caribbean and Atlantic |

**Cohort status**

■ Excluded
■ Included

**Extended Data Fig. 5 | Cartogram-pies indicating the proportion of individuals born in each country (name shown on top of pies) in the XSA cohort.** Pies are placed roughly according to their country's position on a world map. Grey and white squares represent sea and land respectively.

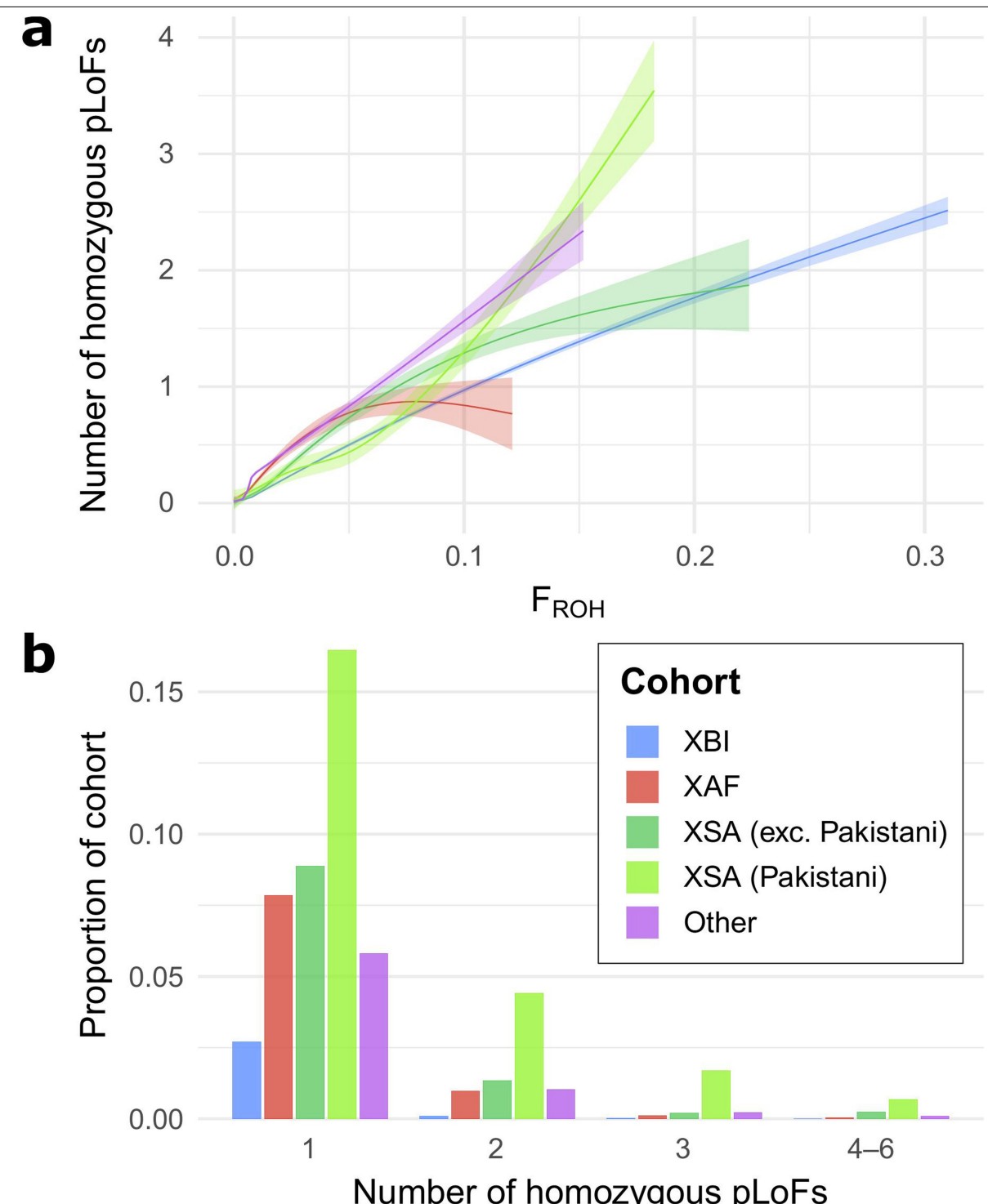

**Extended Data Fig. 6 | Loss-of-function. a**, Correlation between the number of LoF genes per sample and fraction of genome with runs of homozygosity. Shaded region represents 95% confidence interval. **b**, Number of homozygous loss-of-function (LoF) genes per sample. Count of homozygous genes annotated as high impact with frequency <1%. Results are presented for XBI, XAF, XSA excluding individuals self-identified as Pakistani, individuals self-identified as Pakistani from the XSA cohort and Others.

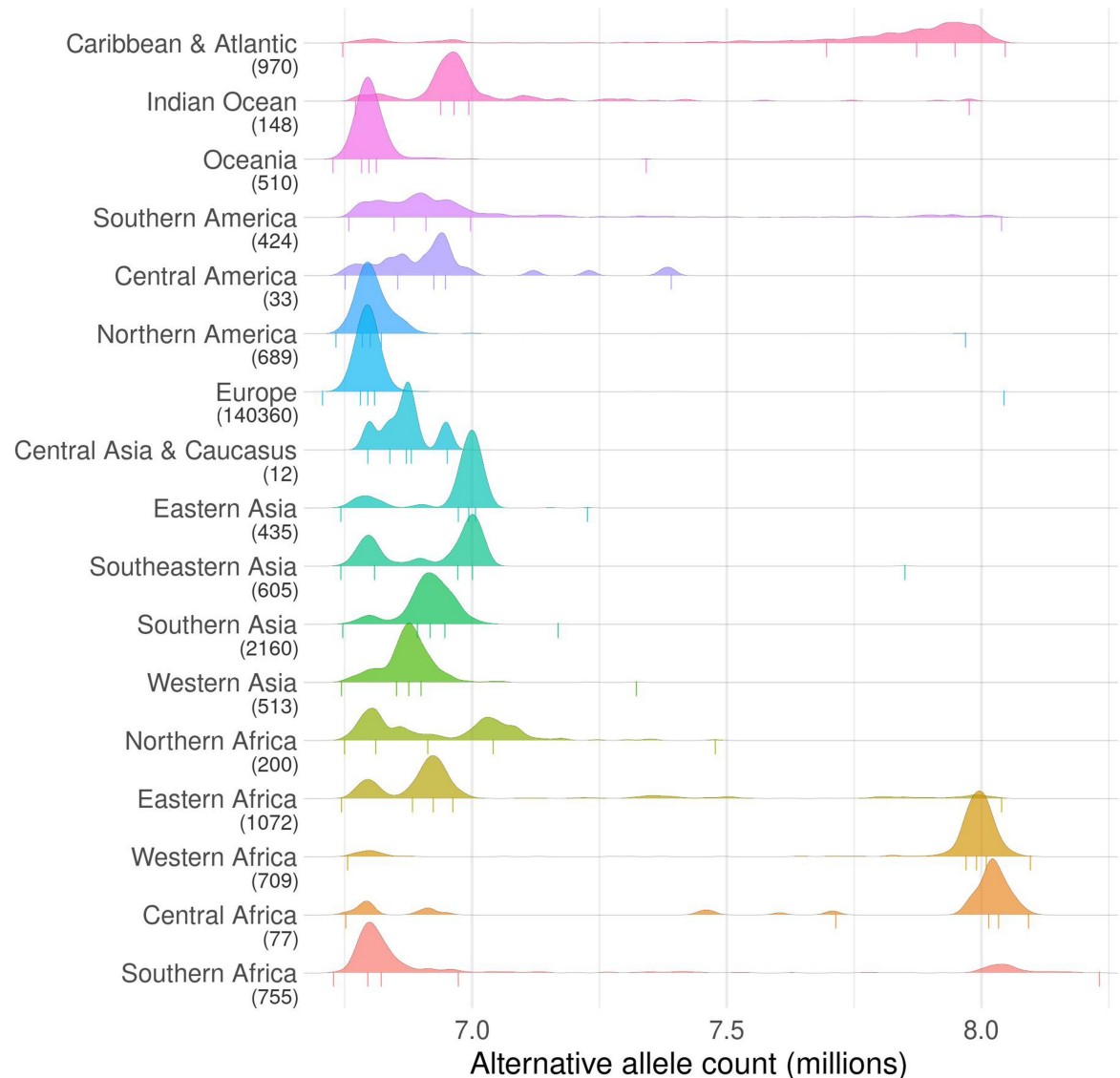

**Extended Data Fig. 7 | Alternative alleles by region.** Numbers in brackets beneath region names indicate count of whole genome sequenced individuals with birthplaces in that region. Assignment of countries to regions is almost identical to the categorization displayed in the cohort cartogram pie figures, with the exception that all European regions are combined into one region in this figure. Vertical lines underneath density curves represent 0th, 25th, 50th, 75th, and 100th percentiles.

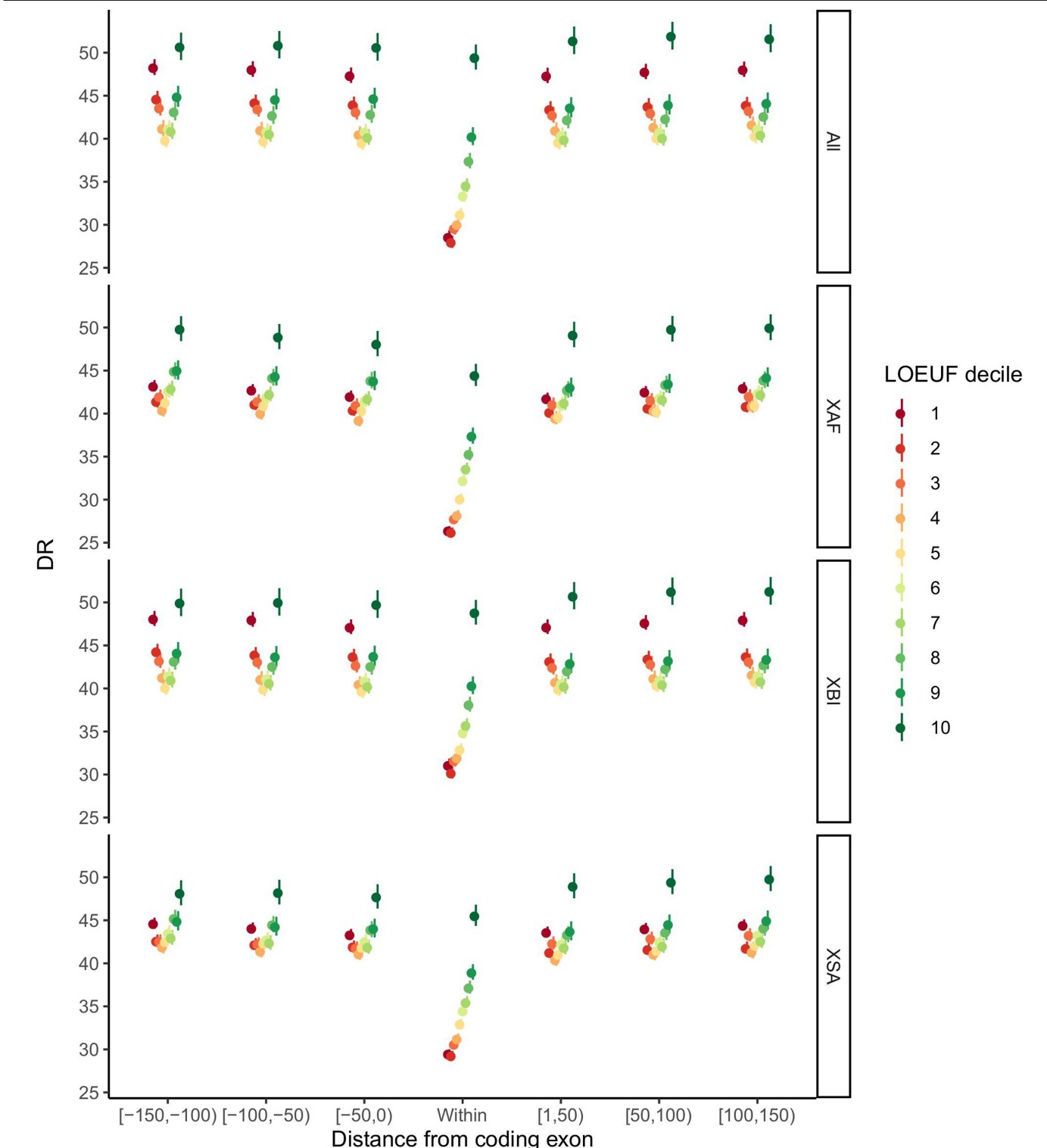

**Extended Data Fig. 8 | DR as a function of distance from coding exon partitioned by LOEUF22 deciles.** Results are shown separately for the overall dataset (All) and the individual cohorts, XBI, XAF and XSA.

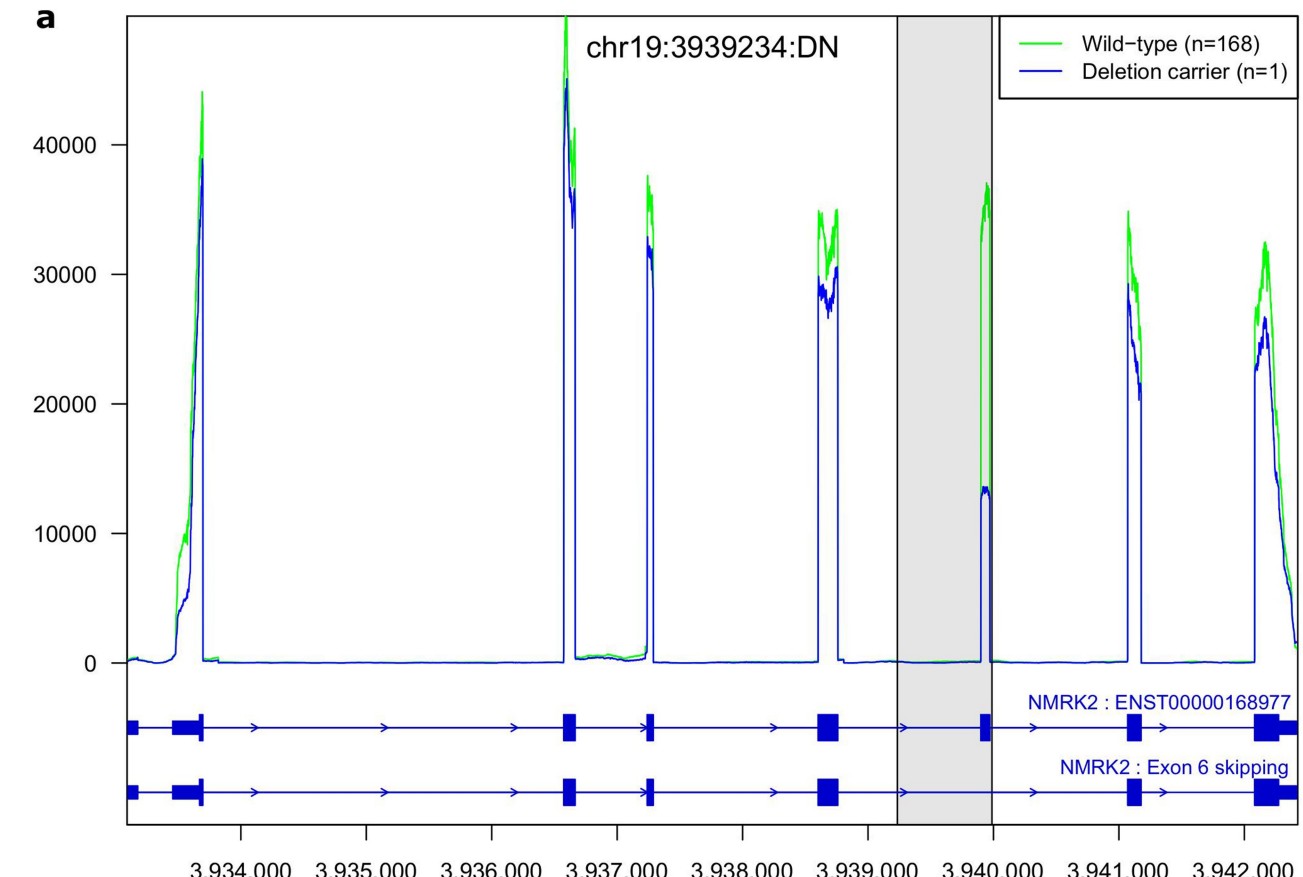

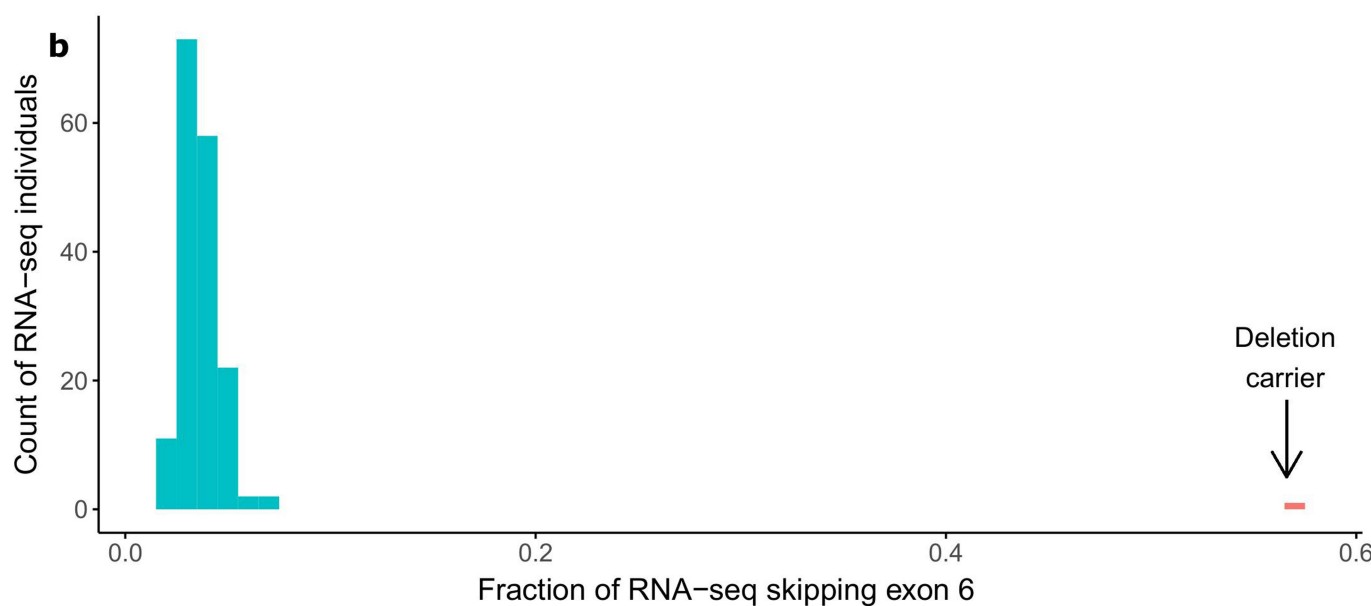

**Extended Data Fig. 9 | RNA analysis of NMRK2. a**, Coverage plot of RNA-sequenced reads from heart tissue from 169 heart tissue samples over the gene NMRK2. One individual is a carrier of a 754bp deletion depicted with gray rectangle that includes exon 6 of NMRK2. The RNA-coverage of the carrier (blue) is lower over exon 6 compared to median coverage of non-carriers (green). Shading marks the deleted region. **b**, Histogram of fraction of RNA-sequenced fragments skipping exon 6 in NMRK2 out of all fragments aligning from the donor site of exon 5 to either acceptor site of exon 6 or exon 7. The median fraction fragments skipping for wild-type individuals is 0.035 and 0.57 for the carrier of the 754bp deletion.

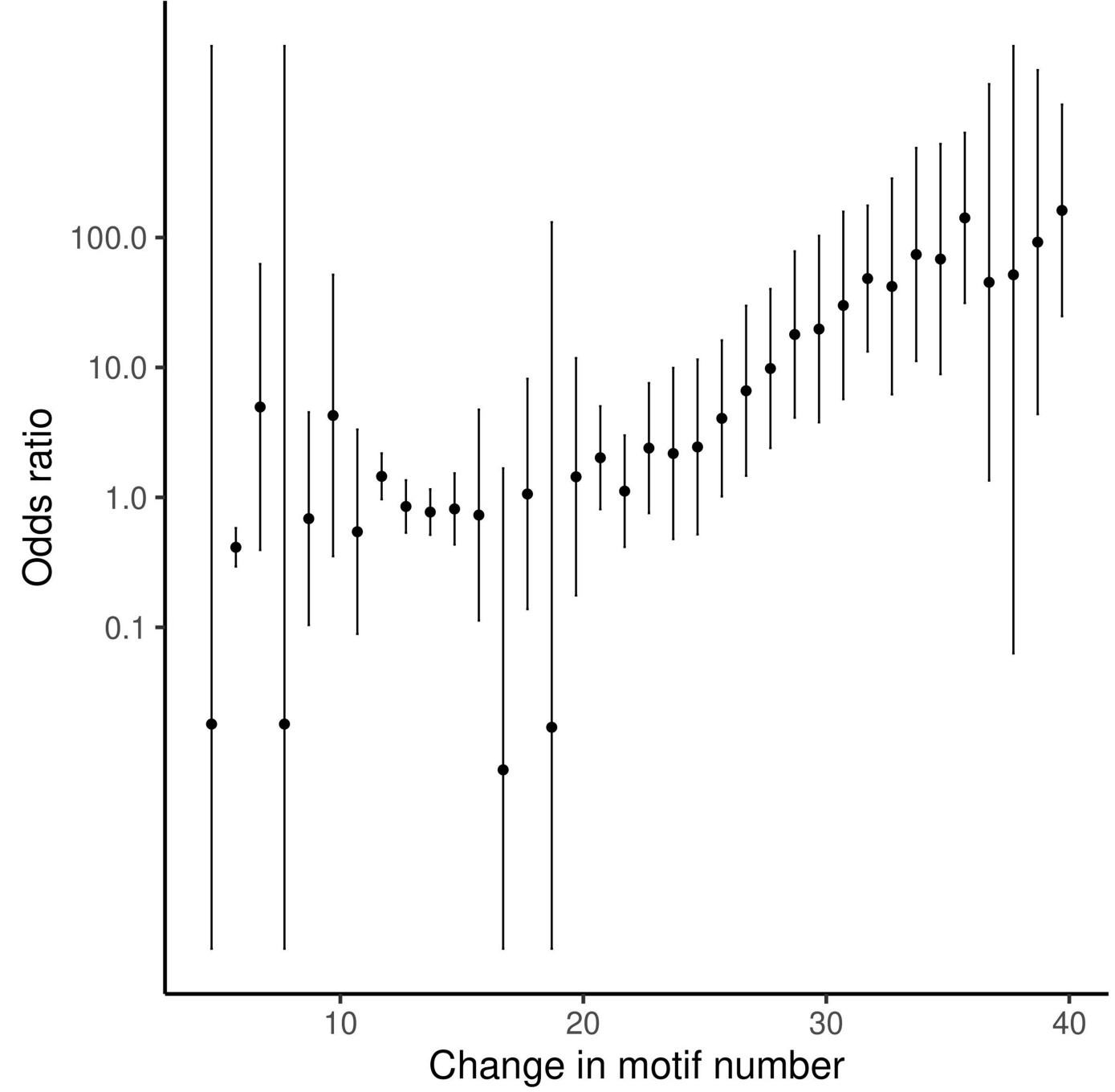

**Extended Data Fig. 10 | Odds ratio for risk of myotonic dystrophy as a function of repeat length in microsatellite at the 3' untranslated region of DMPK.** Carriers of at least 39.7 copies of the microsatellite repeat motif have a 162-fold increased risk of myotonic dystrophy. Error bars represent 95% confidence intervals, n = 431,079.

                           Kari Stefansson

# Reporting Summary

## Statistics

For all statistical analyses, confirm that the following items are present in the figure legend, table legend, main text, or Methods section.

| n/a | Confirmed | |
|---|---|---|
| ☐ | ☒ | The exact sample size (*n*) for each experimental group/condition, given as a discrete number and unit of measurement |
| ☐ | ☒ | A statement on whether measurements were taken from distinct samples or whether the same sample was measured repeatedly |
| ☐ | ☒ | The statistical test(s) used AND whether they are one- or two-sided<br>*Only common tests should be described solely by name; describe more complex techniques in the Methods section.* |
| ☐ | ☒ | A description of all covariates tested |
| ☐ | ☒ | A description of any assumptions or corrections, such as tests of normality and adjustment for multiple comparisons |
| ☐ | ☒ | A full description of the statistical parameters including central tendency (e.g. means) or other basic estimates (e.g. regression coefficient) AND variation (e.g. standard deviation) or associated estimates of uncertainty (e.g. confidence intervals) |
| ☐ | ☒ | For null hypothesis testing, the test statistic (e.g. *F*, *t*, *r*) with confidence intervals, effect sizes, degrees of freedom and *P* value noted<br>*Give P values as exact values whenever suitable.* |
| ☒ | ☐ | For Bayesian analysis, information on the choice of priors and Markov chain Monte Carlo settings |
| ☒ | ☐ | For hierarchical and complex designs, identification of the appropriate level for tests and full reporting of outcomes |
| ☐ | ☒ | Estimates of effect sizes (e.g. Cohen's *d*, Pearson's *r*), indicating how they were calculated |

*Our web collection on statistics for biologists contains articles on many of the points above.*

## Software and code

Policy information about availability of computer code

| Data collection | We used publicly available software URLs are listed below. BamQC (v 1.0.0), https://github.com/DecodeGenetics/BamQC. GraphTyper (v2.7.1), https://github.com/DecodeGenetics/graphtyper. GATK resource bundle (v4.0.12), gs://genomics-public-data/resources/broad/hg38/v0. Svimmer (v0.1), https://github.com/DecodeGenetics/svimmer. popSTR (v2.0), https://github.com/DecodeGenetics/popSTR. Dipcall (v0.1), https://github.com/lh3/dipcall. RTG Tools (v3.8.4), https://github.com/RealTimeGenomics/rtg-tools. bcl2fastq (v2.20.0.422), https://support.illumina.com/sequencing/sequencing_software/bcl2fastq-conversion-software.html. Samtools (v1.9), http://www.htslib.org/. Samblaster (v0.1.24) https://github.com/GregoryFaust/samblaster. |
|---|---|
| Data analysis | We used R (v3.6.0) https://www.r-project.org/ extensively to analyze data and create plots. |

For manuscripts utilizing custom algorithms or software that are central to the research but not yet described in published literature, software must be made available to editors and reviewers. We strongly encourage code deposition in a community repository (e.g. GitHub). See the Nature Portfolio guidelines for submitting code & software for further information.

## Data

Policy information about availability of data

All manuscripts must include a data availability statement. This statement should provide the following information, where applicable:
- Accession codes, unique identifiers, or web links for publicly available datasets
- A description of any restrictions on data availability
- For clinical datasets or third party data, please ensure that the statement adheres to our policy

WGS, genotype data, phased and imputed data can be accessed via the UKB research analysis platform (RAP), https://ukbiobank.dnanexus.com/landing. The

# Field-specific reporting

Please select the one below that is the best fit for your research. If you are not sure, read the appropriate sections before making your selection.

☒ Life sciences      ☐ Behavioural & social sciences      ☐ Ecological, evolutionary & environmental sciences

For a reference copy of the document with all sections, see nature.com/documents/nr-reporting-summary-flat.pdf

# Life sciences study design

All studies must disclose on these points even when the disclosure is negative.

| | |
|---|---|
| Sample size | The UKB has approximately 500,000 samples, a pseudo-random subset of 150,119 of those were sequenced. No statistical analysis was performed to choose sample size. The study presented here is the first data release of the WGS of all 500,000 participants, the choice of sample size was a balance between purported power in association analysis and a consideration of the time from start and finish of the project. All available samples in the Icelandic and Danish replication cohorts were used for replication analysis. |
| Data exclusions | A small subset of individuals withdrew consent during the time of study. |
| Replication | We attempted replication for SNPs when the phenotype information was available and the variant was present and common enough in a replication. We attempted replication for rare variants in menarche and uric acid in Denmark and Iceland respectively. In both cases the association was successfully replicated. |
| Randomization | Samples were pseudorandomly selected among the set of 500,000 samples in the UK biobank and pseudorandomly distributed between the two sequencing centers. |
| Blinding | Investigator were blinded to the randomization. |

# Reporting for specific materials, systems and methods

We require information from authors about some types of materials, experimental systems and methods used in many studies. Here, indicate whether each material, system or method listed is relevant to your study. If you are not sure if a list item applies to your research, read the appropriate section before selecting a response.

## Materials & experimental systems

| n/a | Involved in the study |
|---|---|
| ☒ ☐ | Antibodies |
| ☒ ☐ | Eukaryotic cell lines |
| ☒ ☐ | Palaeontology and archaeology |
| ☒ ☐ | Animals and other organisms |
| ☐ ☒ | Human research participants |
| ☒ ☐ | Clinical data |
| ☒ ☐ | Dual use research of concern |

## Methods

| n/a | Involved in the study |
|---|---|
| ☒ ☐ | ChIP-seq |
| ☒ ☐ | Flow cytometry |
| ☒ ☐ | MRI-based neuroimaging |

## Human research participants

Policy information about studies involving human research participants

| | |
|---|---|
| Population characteristics | Characteristics of the UK biobank have been described in Sudlow, C. et al. UK Biobank: An Open Access Resource for Identifying the Causes of a Wide Range of Complex Diseases of Middle and Old Age. PLOS Med. 12, e1001779 (2015). |
| Recruitment | Recruitment of individuals to the UK biobank has been described in previous studies, the research describes the whole genome sequencing of those samples. |
| Ethics oversight | The North West Research Ethics Committee reviewed and approved UKB's scientific protocol and operational procedures (REC Reference Number: 06/MRE08/65). Data for this study were obtained and research conducted under the UKB applications license numbers 24898 and 68574. |

Note that full information on the approval of the study protocol must also be provided in the manuscript.

