## [Peer Review File · Nature]

Manuscript Title: The sequences of 150,119 genomes in the UK biobank

Reviewer Comments & Author Rebuttals

Reviewer Reports on the Initial Version:

Referee expertise:

Referee #1: GWAS

Referee #2: sequencing, SVs

Referee #3: population genetics

Referees' comments:

Referee #1 (Remarks to the Author):

Page 2, Introduction section, "500,000 largely healthy subjects". How do you define "largely healthy"? Aren't there tens/hundreds of thousands of participants with serious conditions, like autoimmune diseases, COPD, liver disease, cancer, dementia, etc?

Page 2, Introduction section, "an unparalleled opportunity to study sequence diversity". Is this true for rare coding variation and common non-coding variation?

Page 3, Introduction section, "Second, we describe three ancestry-based cohorts within the UKB; with 431,805, 9,633 and 9,252 individuals with British-Irish, African and South Asian ancestries, respectively." How is this a novel finding? Not clear why this is being prominently highlighted, as these ancestral groups are routinely described and analyzed in the UKB. Not sure if it's true that most GWAS use only the White-British subset of UKB. No doubt many do, but many others have used all genetically-confirmed Europeans.

Page 3, Results section, would be useful to start by saying how the 159K individuals were selected for sequencing. Are they a random set of the 500K? If not, what criteria were used to select these individuals?

Page 3, were samples randomly assigned for sequencing to deCODE and Sanger? If not, please describe process. Were any samples sequenced in duplicate, at deCODE and Sanger? How concordant were variant calls? The Methods section describes sequencing at deCODE but not Sanger. Please add the corresponding information for Sanger.

Page 3, "149,960 out of 150,119 individuals could be used for subsequent analysis." Briefly explain why.

Figure, Supplementary Figures and Supplementary Tables are not cited in numerical order. For

example, Fig. S22 and Table S4 are the first cited on page 3, Fig 4d is the main figure cited.

Page 3, the extensive comparison between GATK and GraphTyper call sets is very informative and appreciated. To help readers, I'd suggest briefly highlighting key findings of the comparisons in the main text and adding details about the analyses carried out as footnotes to Supplementary Tables. For example, in Table S4, "GIAB" is not defined, what is the F1 score, how many variants went into this analysis – without these details, it's difficult to interpret results in this Table. Same for Tables S6, S8, S11, S12.

Page 23, Table 1, name used for column 4 ("WES\WGS") is not explicit. Replace with eg. "Unique to WES", and perhaps express as % and not absolute N. Last two cells in bottom row ("100.00%" and "0.00%"), consider replacing with ">99.99%" and "<0.01%".

Page 4, "A clear indication of this is that only (14%) of the highly saturated CpG>TpG variants are singletons". The term "saturated" is not explicit here, and likewise on page 5 ("we see less saturation of", "less saturated"). The saturation ratio is described in the Methods, but it should also be summarized here.

Page 5, "expected number given the nucleotide composition of the window". Briefly summarize how the expected number was derived.

Supplementary Material, page 80, "To our surprise, we saw that a large fraction of the common variants are highly correlated with the sequence center (Table S7), on average of 7.47% and 1.01% of variants for GATK and GraphTyper, respectively." Why was this a surprise? This analysis must be extended to rare variants. The argument that "we won't expect many rare variants would fail the test in general due to lack of sample size" is not convincing at all. In fact, the expectation should be the opposite, that thousands of rare variants should have evidence for batch effects. Please formally compare allele frequencies at all variants between the two sequencing centers (so perform a GWAS of deCODE vs Sanger). Then please provide a Manhattan plot to summarize the results, as well as total number of variants with allele frequency differences between the two centers at e.g. $P < 0.05$, $P < 0.005$, $P < 5e-4$, $P < 5e-8$. This should be done separately for the three different ancestry cohorts.

Supplementary Material, page 91, "500 bp windows where at least 450 bp were considered reliable bps were considered for further analysis.". "Reliable is not defined". Do you mean with coverage >23X? Would also be helpful to indicate the total number of 500 bp windows included in the analysis.

Page 5, "we define FRV as the fraction of variants carried by at most 4 WGS individuals". Why 4 individuals? Why not singletons, or doubletons? And it is difficult to interpret a difference of $FRV = 74.7\%$ (depletion rank <5) vs $FRV = 69.6\%$ (depletion rank >95). Is this really strong evidence that the depletion rank can be used to identify regions under negative selection? For comparison, what is the difference in FRV between eg. stop-lost variants and synonymous variants?

Page 5, "We find that there is a correlation between DR and interspecies scores as measured by GERP33 ($r^2 = 0.0049$, $p = 0.00052$, Fig. 1d)." This shows that there is very little agreement between

the DR and GERP scores. In Fig 4d (not 1d), the middle line (“Mean”) is almost flat across the bulk of the DR distribution. Is this not surprising? Why is there such low agreement and does that not question the validity of the DR score?

Page 5, “Overall, our results indicate that DR can be used to measure negative selection across the entire genome and as such provides a valuable resource for identifying non-coding sequence of functional importance.” Given two comments above, it is not clear to me that this conclusion is justified.

Page 5, “Most GWAS37,39 on the UKB set have been based on a prescribed5 Caucasian subset of 409,559 participants who self-identify as White British.” Is this really true? References 37 and 39 refer to two exome-wide association studies carried out recently, not GWAS. What proportion of published GWAS performed on the UKB cohort used White British as opposed to all Europeans, defined based on genetic clustering with 1000 Genomes samples?

Supplementary Material, page 97, “we defined the cohorts by manually delineating regions in the UMAP latent space that were limited to individuals with British Irish ancestry (XBI, N=431,805), South Asian ancestry (XSA, N=9,633), and African ancestry (XAF, N=9,252).” The manual step seems a bit subjective and probably difficult to reproduce exactly by others.

Page 6, “We crossed GraphTyperHQ variants with exon annotations and found that on average around one in thirty is homozygous for rare”. One in thirty what? Individuals? Exons?

Page 6, “The average number of singletons per individual varies considerably by ancestry (Fig. 1d).” Isn’t this simply because of the difference in total N across ancestries? The explanation for this needs to be clear to readers.

Fig 1e, please add ancestry labels to each plot.

Supplementary Material, page 87, “We refer to a variant as being reliably imputed if leave one out r-squared (L1oR2) of its phasing is greater than 0.5 and imputation info1 was above 0.8.” This needs to be explained better. What exactly is the “r2 of its phasing”? When used to assess imputation performance, r2 is typically calculated based on two sets of genotype calls: (i) set 1, genotypes for variant X obtained through eg. whole-genome sequencing in N individuals that are part of a test set (not included in the reference panel); and (ii) set 2, inferred genotypes for the same variant X and the same N individuals, obtained through imputation using information from eg. array variants only (not including variant X). But this is not what the authors did. Can you please explain step by step what was done to determine how accurately you were able to impute variants identified in whole-genome sequencing of 150K individuals? It is also important to assess accuracy of imputation by (i) phasing WGS data and creating a reference panel using only e.g. 140K individuals (leaving 10K out); (ii) using this new reference panel and phased array data for the remaining 10K individuals to impute WGS variants; and (iii) estimate the r2 between the observed (from WGS) and imputed genotypes for these 10K individuals. This only needs to be done for a chromosome, not the entire genome.

Supplementary Material, Table S14 uses imputation info score >0.5 to identify “imputed variants”,

although other tables (eg. Table S5 use >0.8). This threshold should be standardized across all results.

How were the 16 phenotypes listed in Table S3 specifically selected for analysis?

Page 7, association between rs117919628 and height. As noted, this variant is imputable in UKB at an info score >0.5 using the HRC+1KG+UK10K reference panel. The authors suggest that this represents “low imp info”, but in fact most GWAS (especially very large ones) use an info score >0.3 for imputed variants. And couldn't this variant be imputed more accurately with the TOPMed imputation panel? Lastly, the frameshift variant (rs763014119) has previously been reported to associate with lower height (-0.61 s.d. units, so a larger effect than for rs117919628), through imputation using exome-sequencing data for 50K individuals as a reference panel (PMID 34226706). So the association between GHRH and height does not require whole-genome sequence data to be identified with genome-wide significance through GWAS.

Page 7, associations between rs939016030 (TAC3) and age of menarche, and between rs1383914144 and uric acid. These could be better examples of associations identified through WGS-based imputation, but not other approaches. I note however that the nearest transcript to rs1383914144 is ~ 3.6 Mb away. Please include locuszoom plots for these examples. However, I am concerned that these two associations might be false-positives, see comments below. Were consistent associations observed using WGS data of 150K individuals (ie not imputed data for the remaining 350K individuals)? If so, were the associations consistent between the deCODE and Sanger WGS batches? Have you tried to validate genotype calls from imputation? If these associations with rare imputed variants are to be believed, then replication in an independent study should be provided.

As far as I can tell, the authors provide no details on how robust the GWAS results were for any of the traits tested. In the Methods section, the authors indicate that logistic regression was used for binary traits, I'm assuming linear regression was used for quantitative traits. Is that correct? How did the association analyses account for familial relationships? If what was done was to perform logistic/linear regression and then use LD-score intercepts to correct association test statistics, this is not best-practice and it is not appropriate (and unless I missed it, the authors don't even provide lambdas or LD-score intercepts). If that is the case, then association analyses should be repeated with approaches that account for familial relationships and population substructure while being robust for the analysis of rare variants include REGENIE and BOLT-LMM (for quantitative traits). And were the quantitative traits normalized prior to analysis, appropriately accounting for outliers? Table S3 indicates that some traits were “standardized”, not clear if this means normalized. Please provide Manhattan plots, QQ plots and lambdas (separately for common and rare variants), LD-score intercepts and attenuation ratios for all traits (binary and quantitative) tested. And histograms for the quantitative traits after covariate adjustment.

And in these GWAS, 150K samples had WGS data, with the remaining ~ 350 K having imputed data for WGS variants. How did you handle this difference in data quality between the two datasets? Isn't there an issue whereby phenotypes/diseases that correlate with WGS (150K) vs WGS-imputation (350K) status will result in spurious associations with variants that were in the WGS reference panel

but not imputed very accurately? This is different from what is usually done in GWAS, because typically no samples included in the GWAS are actually part of the reference panel. If thousands of samples were in the reference panel, genotypes for those samples are determined with little error compared to genotypes obtained through imputation.

Regarding the association with uric acid, the authors also note that “A second variant rs1189542743, 4Mb downstream at the end of 1p is strongly correlated ($r^2 = 0.68$) and yields a similar association to uric acid.” This is an extremely large r^2 for two variants that are so far apart (4 Mb). This is highly unusual. Could this not represent a duplication event, or some other structural re-arrangement? This is the centromere of chromosome 1, so presumably difficult to sequence. How many carriers of each variant were included in the analysis? Was the association with uric acid observed in Europeans? If so, any carriers in non-Europeans?

Page 11, you state that “A study of WES from 455K individuals in UK biobank recently reported several examples of associations, including those from gene burden analysis. According to the authors, a majority were either in part previously reported, or could not be replicated. It is noteworthy that none of the associations reported here were found in that comprehensive survey of UKB exome variation.” Where exactly in the Backman et al. study is it mentioned that the majority of associations were previously reported or could not be replicated? The abstract says “Of the signals available and powered for replication in an independent cohort, 81% were confirmed”. To me, this indicates that most were replicated. In the text, Backman et al also state that “As it is not possible to exhaustively describe all novel gene associations”, suggesting that many were not previously reported, no? And why is it noteworthy that the handful of associations reported here were not discovered in the Backman et al. analysis? Weren't the associations that are highlighted in this manuscript of WGS data specifically selected to illustrate examples of associations missed by GWAS and exome-sequencing studies?

Referee #2 (Remarks to the Author):

This paper is the much anticipated report on the variant landscape among 150,110 whole genome sequenced samples from the UK biobank. I expect that the findings from this paper will have an immediate impact on the field, in particular the allele frequencies among diverse cohorts and the DR score. It was good to see that they moved beyond the standard small-variant approach and reported on SVs and micro satellites. Overall the paper is an easy to read and straight-forward presentation of the variants. Interest and impact could be improved by deeper discussions of some of the more novel contributions including DR score and non-coding SVs.

I do have issues and questions regarding the DR score. This section is of high interest to the medical genetics and bioinformatics community. An expanded section that better connects DR to health or other traits would be very useful. The authors could simply follow the basic structure in the gnomAD papers (e.g., Karczewski 2020) with updates based on the larger sample size.

The methods section cites prior work (J Di et al. 2018) as the basic process for DR. In that paper the authors have their own scoring system called context-dependent tolerance score (CDTS). What is the difference between DR and CDTS? How do they compare?

What is the resolution of DR? Coding exons have a mean DR of 28.3. We know from gnomAD (Karczewski 2020) that there is a high degree of variability in the constraint between genes. Haploinsufficient genes are more constrained than olfactory genes and essential genes are more constrained than non-essential genes. Is DR granular enough to capture these differences? Does DR agree with LOUEF (gnomAD's score)? If haploinsufficient genes do have a low DR, are there non-coding regions that have as low or lower DR? What regions have the lowest DR?

The result where DR and GERP do not agree is yet another area that could be explored to increase interest and impact.

Do the DR scores differ when calculated using the different sub-cohorts? We already know that polygenic risk scores are ancestry-specific, but to my knowledge there has not been an exploration of a similar effect in constraint.

It is not clear from the data availability section that the DR scores will be made public.

The SV section gives specific connections between traits and variants that disrupt exons, which is great. An outstanding question in the SV field is what to do with non-coding SVs. The authors discussed associations between non-coding SNVs and traits (e.g., height). From the stats associated with the SVs discussed it seems that the authors did (or could do) a similar trait association using SVs. Were there any associations between non-coding SVs and traits? Even if this answer is no, it would be very interesting.

Referee #3 (Remarks to the Author):

In this article the authors are analyzing the first release of whole genome sequencing (WGS) data from the UK Biobank, representing ~150k individuals in total. Throughout the manuscript the authors draw direct comparisons to what WGS data allows them to discover and analyze in comparison to whole exome sequencing (WES) data and genotype SNP data. In particular the authors look at the genome-wide rare variation they now have access to, they create a new metric to look at potential evidence for purifying selection/functional constraint (Depletion Rank), they compare European and non-European ancestry samples, and they emphasize the importance of both structural variants and microsatellites in gene discovery. This is clearly a unique and exciting dataset as it is the largest collection of WGS data to date. And the authors do discuss what this data brings to the table that has been unavailable to human geneticists up until this point. However, I feel there are multiple areas where this manuscript could be further strengthened. This includes better explaining some of the points and analyses already shown as well as going deeper in some of the sections. I outline both my major and minor feedback below.

Major Points:

a) Near the end of page 4, the authors discuss the qualities of rare variation and de-novo mutations in their data. While this paragraph provides interesting results, I feel it could use some more context for a broader audience. What is the exact importance of trying to identify the relative rate of recurrent mutations versus IBD? The authors state "Due to the scale of the UKB WGS data, an observation of the same allele in unrelated individuals does not always imply identity by descent."

This is useful, but I think they should include an additional sentence or two explaining how this distinction is important for the type of analyses we do in human and population genetics. Additionally for this paragraph, the authors show: “Out of the 194,687 Icelandic DNMs we find 53,859 (27.7%) in the UKB set...”; it is unclear to me how to interpret this result. Is an overlap of 27.7% expected? If the authors could provide some additional context, whether analyses, simulations, or just perspective, I think that would also be helpful for the audience here.

b) In the middle of page 5, I find the Depletion Rank (DR) analysis to be particularly interesting. I do have a few questions however. One is, at the end of the section, the authors conclude that DR will help with “...identifying non-coding sequence of functional importance.” Is there a way for the authors to more directly show that this is the case? Do they find enrichment for any type of regulatory architecture amongst the regions that have low DR’s? I think since there are tools and methods available now for combining functional genomic information with GWAS hits, trying to run some of these analyses in the regions of low DR could be a way to explicitly show the utility of DR. Specifically, by focusing on regions of low DR, you might alleviate the multiple testing burden and have a more liberal Bonferroni-corrected p-value threshold, thus allowing you to designate more regions as significant for whatever test or metric you end up using. Alternatively, do any of the novel associations that the authors highlight later in the paper occur in the lowest DR regions? That would be a nice connection to bring up if so.

One other question is whether the authors have looked at other commonly used metrics of purifying selection, like Tajima’s D or the ExtRaINSIGHT method from the Dukler et al. paper they cite in this section. The authors did compare their DR metric to GERP values, but as they point out this metric uses multi-species comparisons. There are metrics for identifying purifying selection within species as well. The authors could also compare their metric against genome-wide measures of ‘deleteriousness’ as well, such as CADD, Eigen, or LINSIGHT.

Lastly, is there any reason why these analyses were not conducted in the non-European ancestry populations? I think that could be potentially very interesting, since you might find evidence of particularly recent purifying selection – or at least regional differences in purifying selection.

c) At the top of page 7 in the imputation section, the authors state: “It is thus likely that the UKB reference panel provides the best available option for imputing genotypes into population samples from Africa and South Asia.” This is a statement that can be supported by additional analyses, so I am unsure why they were not conducted. Unless I am missing something, only one reference panel was used. To make a statement about how this reference panel performs for non-European ancestry populations versus other reference panels, why not explicitly conduct imputation runs using some of the other reference panels out there, like the 1000G, CAAPA, or TOPMed reference panels? If this new, UKB WGS-based reference panel was indeed better for certain non-European ancestry populations, that would be an impactful finding. But, as far as I can tell, this comparison has not actually been conducted yet.

d) For both the new associations highlighted with the structural variants and microsatellites, it is certainly interesting and important to emphasize that WGS data allows you to better analyze these classes of variants. However, in each section, I think it would strengthen the association results to do some additional follow-up to further support causality. Is there any evidence of any regulatory architecture being present nearby these variants, such as histone mark, methylation, or expression

QTLs? If so it might be possible to conduct some colocalization analyses. Are any of these genes present in OMIM or are previous drug targets? Is there any possibility to do a replication analysis for any of these results, such as what Backman et al. 2021 Nature did with the UKB WES and DiscovEHR data? Could the gnomAD WGS data be useful here?

I think the use of NMRK2 and the right atrium heart RNA-seq data is a great start, but are there any links to a relevant phenotype that can be shown with that data? I'm not sure how impactful it is to use that data just to validate the presence of the deletion; connecting the deletion, its association with age at menarche, and its expression in the heart would certainly be a very interesting finding. Overall, I think trying to take some of these associations 'one step' further in terms of showing causality would be very helpful, whichever direction in particular the authors think would be best. Both the Backman et al. 2021 paper and the Barton et al. 2021 Nature Genetics UKB WES imputation paper may be useful references for comparison.

e) As a consideration for a general, additional area of analyses that could help increase the impact of the manuscript, I am wondering about comparing some of the current results versus what happens when you imitate 'low pass sequencing' on the original data. In other words, purposely downsampling your reads and rerunning your SNP discovery pipeline. There is general interest in low pass WGS as a cheaper alternative for a variety of purposes in clinical genomics, and the UKB WGS data seems like a great opportunity to compare how differences in coverage can affect analytical outcomes. This may be a non-trivial set of additional analyses, but I think further exploring analyses that can only be done with this dataset, such as direct comparison of analyses using different levels of sequence coverage, would be great for this manuscript.

Minor Points:

a) At the end of page 3, "As GraphTyper provided the more accurate genotype calls the GraphTyper genotypes were used...", could some summary performance metric be included to show how much more accurate GraphTyper is? I appreciate there is material in the supplement that is referenced, but I think this is a useful result to support directly in the main text some way.

b) At the beginning of page 4, I think it could be worthwhile to spell out ACMG and also explain what an 'actionable genotype' is. Since this is intended for Nature, the audience is going to be more broad than the typical genomics community that reads these papers. I'm not sure if it should be assumed that the audience will exactly understand the point being made in this paragraph. To that point additionally, it could also be worthwhile to include an additional sentence or two explaining the impact of identifying a greater number of ACMG genes. The methods section provides a bit more detail on this analysis. Moving some of that to the main text might help.

c) In the middle of page 4, the authors show "...Even inside of coding exons currently curated by Encode, we estimate that 10.7% of variants are missed by WES". Is this simply poor, uneven coverage in the WES data? Is this entirely rare variants? This seems like a somewhat surprising result for at least the coding exons – I think some further analysis or at least hypothesizing on what is going on here would be appreciated.

d) Near the end of page 5 in the DR section, the authors highlight a correlation analysis between their DR metric and GERP, but the r^2 value of their analysis is only .0049. While technically significant with a p-value of .00052, I wonder how meaningful this result is with an r^2 below .01. Is there a typo here? I'm not sure if it's fair to say that there is indeed correlation between the two metrics when the value is so close to 0, even if significant. Also I believe the authors are citing Figure 4d, not 1d, in this section as well.

e) In the caption of Figure 3, I think it would be useful to explain here how 3b and 3c show that differences in singletons is likely due to the geographic sampling disparities between northern and southern Britain.

f) Near the top of page 7, the authors state: "We found a number of clinically important variants that can now be imputed from the dataset." Is this statement more emphasizing the importance of the original data or the reference panel created from the data? If it's the latter, has this statement been confirmed by imputing the current dataset with other reference panels? Is it the case that you do not recover most of the same variants or that imputation quality scores are significantly worse? I think some more details here would be useful.

g) The authors can also cite the following, more recent height genetics paper to support their discovery of a large effect height allele: <https://www.nature.com/articles/s41586-020-2302-0>

h) In the structural variants section, the authors state the following: "The deletion and rs147068659 are correlated ($r^2 = 0.67$), after conditional analysis the deletion remains significant ($p = 6.4 \times 10^{-8}$) whereas rs147068659 does not ($p = 0.39$), indicating the deletion is causal." The analyses presented suggest the deletion may be the /lead/ variant for this signal. However, association and conditional analyses do not provide evidence for causality. The use of causal here is inaccurate.

i) For all association analyses conducted, was sequencing site used as a covariate? It is mentioned that roughly two thirds of the samples were sequenced at deCODE Genetics and the rest were sequenced at the Wellcome Trust Sanger Institute. I think at least conducting some sensitivity analyses to check this would be good if that hasn't been done yet.

Author Rebuttals to Initial Comments:

Reviewer #1

Page 2, Introduction section, “500,000 largely healthy subjects”. How do you define “largely healthy”? Aren’t there tens/hundreds of thousands of participants with serious conditions, like autoimmune diseases, COPD, liver disease, cancer, dementia, etc?

Response: The statement was inaccurate, we have rephrased this to: The UK biobank (UKB) documents phenotypic variation across 500,000 subjects across the United Kingdom, with a healthy volunteer bias.

Page 2, Introduction section, “an unparalleled opportunity to study sequence diversity”. Is this true for rare coding variation and common non-coding variation?

Response: Researchers have been able to study sequence diversity of rare coding and common non-coding variants using WES and chip array data. The new study allows us to expand this to rare coding variants, a set of variants 40 times larger than have been studied previously on the same set of individuals.

Page 3, Introduction section, “Second, we describe three ancestry-based cohorts within the UKB; with 431,805, 9,633 and 9,252 individuals with British-Irish, African and South Asian ancestries, respectively.” How is this a novel finding? Not clear why this is being prominently highlighted, as these ancestral groups are routinely described and analyzed in the UKB. Not sure if it’s true that most GWAS use only the White-British subset of UKB. No doubt many do, but many others have used all genetically-confirmed Europeans.

Response: We are not claiming a conceptual novelty here, there are however noteworthy features in how our cohorts are defined, such as the fact that we include Irish individuals along with individuals who have most of their ancestry from the UK, and our use of UMAP. We believe that it was necessary to make this clear in the manuscript as many of our analyses are stratified by the three cohorts. In this context, it is important to bear in mind that this study represents a major expansion in the number of publicly available WGS from diverse ancestries (especially deeply and consistently phenotyped WGS samples) and our results are the first to describe this set. Please also see our related responses below.

Page 3, Results section, would be useful to start by saying how the 159K individuals were selected for sequencing. Are they a random set of the 500K? If not, what criteria were used to select these individuals?

Response: This was previously stated in the supplement: “Samples >12 ng/μl will be selected by UK Biobank using its picking algorithm which ensures pseudo-randomisation of recruitment centers and collection times across batches, to avoid potential batch effects.” We have now noted in the results section that the selection was pseudorandom.

Page 3, were samples randomly assigned for sequencing to deCODE and Sanger? If not, please describe process. Were any samples sequenced in duplicate, at deCODE and Sanger? How

concordant were variant calls? The Methods section describes sequencing at deCODE but not Sanger. Please add the corresponding information for Sanger.

Response: The samples were pseudo-randomly assigned between the two sequencing centers. We have updated the description of the sequencing such that it covers the protocol at both centers.

Page 3, “149,960 out of 150,119 individuals could be used for subsequent analysis.” Briefly explain why.

Response: This difference in the number of individuals is only 0.1%, so we were not concerned that this would impact our analysis. We had previously stated that these individuals withdrew consent. Upon further inspection we realized this was not entirely correct. 11 of the samples had withdrawn consent and 13 samples were sequenced in duplicate. The remaining 135 samples were missing from our analysis as our process of linking sequencing data with phenotype data relied on a list of microarray typed individuals. Our list of microarray typed individuals was not current and we believe that these individuals have in fact been microarray typed and could most likely be included in analysis performed by other researchers.

Figure, Supplementary Figures and Supplementary Tables are not cited in numerical order. For example, Fig. S22 and Table S4 are the first cited on page 3, Fig 4d is the main figure cited.

Response: We have made this modification.

Page 3, the extensive comparison between GATK and GraphTyper call sets is very informative and appreciated. To help readers, I’d suggest briefly highlighting key findings of the comparisons in the main text and adding details about the analyses carried out as footnotes to Supplementary Tables. For example, in Table S4, “GIAB” is not defined, what is the F1 score, how many variants went into this analysis – without these details, it’s difficult to interpret results in this Table. Same for Tables S6, S8, S11, S12.

Response: Thank you. We have added the clarifications suggested by the reviewer to the manuscript. F1 score is the harmonic mean of sensitivity and precision. We have added the definition of F1, GIAB (Genome in a bottle) and the number of variants in the GIAB set to table S4. It is not clear what problem the reviewer sees with table S6 but we have now added the list of the 500 regions tested. We have added the total number for table S8. For tables S11A and B we have added a column with the number of variants tested. For table S12 we have added the number of variants in WES at the given frequency thresholds. We have added a brief summary of the key findings of the comparisons to the end of page 3.

Page 23, Table 1, name used for column 4 (“WES\WGS”) is not explicit. Replace with eg. “Unique to WES”, and perhaps express as % and not absolute N. Last two cells in bottom row (“100.00%” and “0.00%”), consider replacing with “>99.99%” and “<0.01%”.

Response: We have made the changes suggested.

Page 4, “A clear indication of this is that only (14%) of the highly saturated CpG>TpG variants are singletons”. The term “saturated” is not explicit here, and likewise on page 5 (“we see less saturation

of”, “less saturated”). The saturation ratio is described in the Methods, but it should also be summarized here.

Response: We have added the definition of saturation to results.

Page 5, “expected number given the nucleotide composition of the window”. Briefly summarize how the expected number was derived.

Response: We have added the description of the expectation.

Supplementary Material, page 80, “To our surprise, we saw that a large fraction of the common variants are highly correlated with the sequence center (Table S7), on average of 7.47% and 1.01% of variants for GATK and GraphTyper, respectively.” Why was this a surprise? This analysis must be extended to rare variants. The argument that “we won’t expect many rare variants would fail the test in general due to lack of sample size” is not convincing at all. In fact, the expectation should be the opposite, that thousands of rare variants should have evidence for batch effects. Please formally compare allele frequencies at all variants between the two sequencing centers (so perform a GWAS of deCODE vs Sanger). Then please provide a Manhattan plot to summarize the results, as well as total number of variants with allele frequency differences between the two centers at e.g. $P < 0.05$, $P < 0.005$, $P < 5e-4$, $P < 5e-8$. This should be done separately for the three different ancestry cohorts.

Response: As the two sequencing centers followed a very similar sequencing protocol this was a surprise to us, although perhaps it should not have been. The statement “we won’t expect many rare variants would fail the test in general due to lack of sample size” was meant to convey that we have less power to detect batch effects for rare variants. We have now made this statement more accurate.

The reviewer is correct that batch effects may also exist for rare variants and we have added the experiments suggested by the reviewer, reporting the number of variants that associate with the sequencing center as a function of frequency and the p-value thresholds suggested. As expected, we are more likely to reject the null hypothesis of the frequency being the same between sequencing centers for common markers than rare ones.

The GWAS results presented in the manuscript are based on imputed data, for completeness we present the sequencing center batch effect results both for the imputed and directly sequenced genotypes. Results are presented with and without filtering for AAScore and imputation info, as expected there is greater evidence of batch effects in the unfiltered set. A much smaller number of variants shows batch effects in the filtered dataset, but still more than expected by chance.

Manhattan plots are not informative for the batch effects. A large number of variants that associate with the sequencing center are detected and these markers are largely randomly distributed across the genome.

Supplementary Material, page 91, “500 bp windows where at least 450 bp were considered reliable bps were considered for further analysis.”. “Reliable is not defined”. Do you mean with coverage $> 23X$? Would also be helpful to indicate the total number of 500 bp windows included in the analysis.

Response: The term “reliable bps” is defined at the top of the same page; “The sequence coverage at each bp in GRCh38 was computed for each of 1,000 randomly selected individuals. At each bp we then computed the mean and s.d. of coverage across the 1,000 individuals. Bps with mean coverage at least 20 and s.d. of coverage at most 12 were considered reliable bps.”

We have added the number of 500bp windows analyzed, the number of windows corresponds to approximately 90% of the genome.

Page 5, “we define FRV as the fraction of variants carried by at most 4 WGS individuals”. Why 4 individuals? Why not singletons, or doubletons? And it is difficult to interpret a difference of FRV=74.7% (depletion rank <5) vs FRV=69.6% (depletion rank >95). Is this really strong evidence that the depletion rank can be used to identify regions under negative selection? For comparison, what is the difference in FRV between eg. stop-lost variants and synonymous variants?

Response:

Around half of the variants are singletons and 96.6% of markers have frequency below 0.1%. It is clear that a cutoff needs to have a small number of carriers to be informative.

We considered other cutoffs in Fig S3, the shape of the FRV curve on the DR quantiles is similar for the different frequency cutoffs. We appreciate the excellent suggestion of the reviewer to provide a yardstick to the reader by comparing the FRV of DR quantiles to functional categories of coding variants. We added the following sentence to the main text and horizontal dashed lines to the DR/FRV figure corresponding to the FRVs of coding variants.

“We used FRV of annotated coding variants as a reference (Fig. 1f), and we find that the 1% of the DR has a comparable FRV to a missense mutation (75.5%).”

Page 5, “We find that there is a correlation between DR and interspecies scores as measured by GERP33 ($r^2 = 0.0049$, $p = 0.00052$, Fig. 1d).” This shows that there is very little agreement between the DR and GERP scores. In Fig 4d (not 1d), the middle line (“Mean”) is almost flat across the bulk of the DR distribution. Is this not surprising? Why is there such low agreement and does that not question the validity of the DR score?

Response:

GERP score is based on a multiple alignment of 36 mammalian species, everything from egg-laying platypus to humans. Where negative selection acts similarly across the mammalia for the last 187 million years (PMID 33408411), sequence conservation in the multiple alignment is expected to be highly congruent with the GERP score. We find that the highly conserved regions at the inter-species level are also conserved within human populations as measured by the blue line in figure 1g. We find a better correlation between DR and a GERP score within exons ($R^2=0.0498$) than outside them ($R^2=0.0012$). This indicates that the negative selection acting outside the coding exons is weaker or transient across the mammalian species compared to the one acting on the coding exons, supporting the notion that there is substantial value in DR outside the coding exons to detect human specific negative selection.

For human and primate specific elements under selection the GERP score is underpowered to detect these elements based on population genetics simulations (PMID 32469868). To construct an assay of human specific negative selection we fetched the loss-of-function observed/expected upper bound fraction (LOEUF) derived by the gnomAD consortium, and modeled LOEUF as function of the average GERP score in the exon with a generalized additive model. We then extracted the residuals from this fit, which we refer to as LOEUF|GERP, the residuals should be a metric for human specific conservation for coding exons as the extent of mammalian sequence conservation is regressed out.

We find a positive correlation between the LOEUF residuals and DR (Pearson's $r=0.29$; $p<2.2e-16$), and LOEUF|GERP residuals and DR (Pearson's $r=0.16$; $p<2.2e-16$). The latter result indicates that the DR score measures human specific selection and provides value beyond mammalian sequence conservation within the coding exons. The construction of the DR is agnostic to the annotation of the coding exons, therefore this result highlights the usefulness of the DR score as it is derived for the entire genome accessible by short reads. We incorporated these results into the main text and added text to better describe the relationship between GERP and DR.

Page 5, "Overall, our results indicate that DR can be used to measure negative selection across the entire genome and as such provides a valuable resource for identifying non-coding sequence of functional importance." Given two comments above, it is not clear to me that this conclusion is justified.

Response: We believe that our response to the two comments and the additions made to the text should make this relationship more clear.

We have now also added the FRV of coding variants as horizontal lines to Figure 1D to give the reader better context of the FRV of the DR percentiles. Further, we expanded the GERP analysis and describe better the relationship between DR and GERP which is greatest in highly conserved sequence among all mammals, such as the coding sequence. We demonstrate the correlation between the LOEUF|GERP residuals measuring human specific sequence conservation and DR (Pearson's $r=0.16$; $p<2.2e-16$). The codon table was not used in the creation of DR score, despite this we find a correlation between LOEUF|GERP residuals and DR rank. The agnostic nature of DR score is key to exploring sequence conservation in the non-coding genome as the non-coding genome has not been as well annotated as the coding part of the genome. Therefore our conclusion is justified as our results demonstrate that DR allows assessment of the sequence conservation of the non-coding genome.

Page 5, "Most GWAS37,39 on the UKB set have been based on a prescribed5 Caucasian subset of 409,559 participants who self-identify as White British." Is this really true? References 37 and 39 refer to two exome-wide association studies carried out recently, not GWAS. What proportion of published GWAS performed on the UKB cohort used White British as opposed to all Europeans, defined based on genetic clustering with 1000 Genomes samples?

Response: Thank you, the statement was inaccurate and we have adjusted accordingly.

As a simple experiment to estimate the proportion that used White British individuals as opposed to all Europeans, we conducted a survey of the first 50 results on Google Scholar for "association uk

biobank". 26 of those reported at least one valid GWAS-like analysis, 13 of these were definitely conducted on some subset of specifically White British individuals. The other 13 studies included individuals who were not White British, but nearly all of these were conducted only on (white) Europeans. The exceptions were 2 papers that used 4 ancestry cohorts, and 1 that did not account for ancestry in any way at all.

Supplementary Material, page 97, "we defined the cohorts by manually delineating regions in the UMAP latent space that were limited to individuals with British Irish ancestry (XBI, N=431,805), South Asian ancestry (XSA, N=9,633), and African ancestry (XAF, N=9,252)." The manual step seems a bit subjective and probably difficult to reproduce exactly by others.

Response: We agree: genetic variation in human populations is continuous and can only be feasibly described using approximations (PMID: 32150538), so partitioning by ancestry inevitably requires some degree of subjective choice in where to place thresholds. Additionally, the methods we employed (ADMIXTURE and UMAP) are partly stochastic, and ensuring perfect reproducibility by specifying, for instance, all the random seeds used at each stage, and the exact batching of samples for ADMIXTURE, does not seem practical, regrettably. We believe that Fig. S30 [ADMIXTURE assignments shown on UMAP] and S28 [cohort boundaries on UMAP] provide enough information to readers to allow them to replicate our general approach and obtain qualitatively equivalent results, and the summaries in Fig. S29 [ADMIXTURE ancestry summaries by cohort] and S6-8 [birthplaces by cohort] should allow them to compare the outcome of their implementation with ours.

We note that we do provide the mapping of the UKB cohorts on the UKB research analysis platform (RAP).

Page 6, "We crossed GraphTyperHQ variants with exon annotations and found that on average around one in thirty is homozygous for rare". One in thirty what? Individuals? Exons?

Response: We have corrected this statement, which refers to one in thirty individuals.

Page 6, "The average number of singletons per individual varies considerably by ancestry (Fig. 1d)." Isn't this simply because of the difference in total N across ancestries? The explanation for this needs to be clear to readers.

Response: This has to do with the sampling from these cohorts and their genetic diversity. We have clarified that this has to do with the sampling in the text, by adding the qualifier "Largely due to the number of individuals sampled".

Fig 1e, please add ancestry labels to each plot.

Response: These have been added.

Supplementary Material, page 87, "We refer to a variant as being reliably imputed if leave one out r-squared (L1oR2) of its phasing is greater than 0.5 and imputation info1 was above 0.8." This needs to be explained better. What exactly is the "r2 of its phasing"? When used to assess imputation performance, r2 is typically calculated based on two sets of genotype calls: (i) set 1, genotypes for variant X obtained through eg. whole-genome sequencing in N individuals that are part of a test set

(not included in the reference panel); and (ii) set 2, inferred genotypes for the same variant X and the same N individuals, obtained through imputation using information from eg. array variants only (not including variant X). But this is not what the authors did. Can you please explain step by step what was done to determine how accurately you were able to impute variants identified in whole-genome sequencing of 150K individuals? It is also important to assess accuracy of imputation by (i) phasing WGS data and creating a reference panel using only e.g. 140K individuals (leaving 10K out); (ii) using this new reference panel and phased array data for the remaining 10K individuals to impute WGS variants; and (iii) estimate the r^2 between the observed (from WGS) and imputed genotypes for these 10K individuals. This only needs to be done for a chromosome, not the entire genome.

Response: Thank you. We have now included the definition of leave-one-out- r^2 (L1oR2) in the text. We phase each individual and impute without considering the genotypes of the individual in question and then compute the r^2 between this imputation and the original genotype calls. This ensures that the imputed genotypes of the sequenced samples agree with their original genotypes. The measure provided is quite similar to the one suggested by the reviewer, instead of withholding a subset of the samples each sample is withheld in turn.

The imputation information (imp info) measures the ratio of the variance of imputed expected allele counts and the variance of the actual allele count and is described in Gudbjartsson 2015, cited in our study. This measure extends to the genotypes in the samples that have not been sequenced.

Supplementary Material, Table S14 uses imputation info score >0.5 to identify “imputed variants”, although other tables (eg. Table S5 use >0.8). This threshold should be standardized across all results.

Response: We have made the change requested.

How were the 16 phenotypes listed in Table S3 specifically selected for analysis?

Response: We performed analysis of a number of phenotypes in the UKB and we selected examples of associations that could not be easily observed in whole exome data or through imputation.

Page 7, association between rs117919628 and height. As noted, this variant is imputable in UKB at an info score >0.5 using the HRC+1KG+UK10K reference panel. The authors suggest that this represents “low imp info”, but in fact most GWAS (especially very large ones) use an info score >0.3 for imputed variants. And couldn't this variant be imputed more accurately with the TOPMed imputation panel? Lastly, the frameshift variant (rs763014119) has previously been reported to associate with lower height (-0.61 s.d. units, so a larger effect than for rs117919628), through imputation using exome-sequencing data for 50K individuals as a reference panel (PMID 34226706). So the association between GHRH and height does not require whole-genome sequence data to be identified with genome-wide significance through GWAS.

Response: The TOPMed imputation panel had not been made available to us when we submitted the manuscript. Our claim that the marker would fail imputation info quality thresholds was inaccurate as these thresholds are defined differently between studies. We now state that the marker may fail imputation info thresholds. We still believe that our statement that the association has not been reported previously is accurate and an interesting application of our dataset.

In the original submission we had noted that rs763014119 is also observed in the same gene. We have now added a citation to PMID 34226706 for this association. This marker is uncorrelated with rs117919628 and consequently our marker represents a novel association.

Page 7, associations between rs939016030 (TAC3) and age of menarche, and between rs1383914144 and uric acid. These could be better examples of associations identified through WGS-based imputation, but not other approaches. I note however that the nearest transcript to rs1383914144 is ~3.6 Mb away. Please include locuszoom plots for these examples. However, I am concerned that these two associations might be false-positives, see comments below. Were consistent associations observed using WGS data of 150K individuals (ie not imputed data for the remaining 350K individuals)? If so, were the associations consistent between the deCODE and Sanger WGS batches? Have you tried to validate genotype calls from imputation? If these associations with rare imputed variants are to be believed, then replication in an independent study should be provided.

Response: We have added locuszoom and replication results for both of these associations, in Denmark for TAC3 and in Iceland for Uric acid and gout.

rs939016030 (TAC3) is identical in the raw and imputed genotypes ($r^2 = 0.999993$). Similarly the Uric Acid markers are nearly identical in raw and imputed genotypes (rs1383914144, $r^2 = 0.998$, rs1189542743, $r^2 = 0.997$). We have added a supplementary table showing the correlations between the markers in the imputed dataset and the raw genotypes for all the markers presented here. We have added a supplementary table where we test all variants presented for sequencing batch effects.

Please see below our response below regarding the reviewer's concerns specifically regarding the placement of the Uric Acid markers.

As far as I can tell, the authors provide no details on how robust the GWAS results were for any of the traits tested. In the Methods section, the authors indicate that logistic regression was used for binary traits, I'm assuming linear regression was used for quantitative traits. Is that correct? How did the association analyses account for familial relationships? If what was done was to perform logistic/linear regression and then use LD-score intercepts to correct association test statistics, this is not best-practice and it is not appropriate (and unless I missed it, the authors don't even provide lambdas or LD-score intercepts). If that is the case, then association analyses should be repeated with approaches that account for familial relationships and population substructure while being robust for the analysis of rare variants include REGENIE and BOLT-LMM (for quantitative traits). And were the quantitative traits normalized prior to analysis, appropriately accounting for outliers? Table S3 indicates that some traits were "standardized", not clear if this means normalized. Please provide Manhattan plots, QQ plots and lambdas (separately for common and rare variants), LD-score intercepts and attenuation ratios for all traits (binary and quantitative) tested. And histograms for the quantitative traits after covariate adjustment.

Response: Thank you. The description of the association testing was inadequate, we have updated the description, we follow the methodology of BOLT-LMM. We have added the requested information for the phenotypes tested, according to the type of variant reported (e.g., only

structural variant Manhattan plot and QQ plot for phenotypes with SV associations). We have indeed performed inverse-normal transformation after obtaining residuals from a generalized additive model (gam) with year of birth, sex and 20 (XBI, XAF) or 45 (XSA) principal components. Histograms for inverse-normal transformed values were added to the Supplementary Figures and a more detailed description was added to Supplementary Table S23.

We correct our associations using LD-score regression intercepts obtained from a subset of 1.1M independent variants. The attenuation ratios obtained are comparable to the original ratios reported by Loh *et al.* (PMID: 29892013) using the same methods. We observe a deflation of rare variants in comparison with common variants for all phenotypes. Unadjusted lambdas (i.e., before LD score regression intercept correction) using all variants show a well-behaved pattern, with max $\lambda=1.107$ for standing height. However, we do notice an underinflation in case/control phenotypes with small numbers of cases, such as hereditary ataxia and myotonic dystrophy. This effect, however, only reduces significance levels and does not lead to false positives.

And in these GWAS, 150K samples had WGS data, with the remaining ~350K having imputed data for WGS variants. How did you handle this difference in data quality between the two datasets? Isn't there an issue whereby phenotypes/diseases that correlate with WGS (150K) vs WGS-imputation (350K) status will result in spurious associations with variants that were in the WGS reference panel but not imputed very accurately? This is different from what is usually done in GWAS, because typically no samples included in the GWAS are actually part of the reference panel. If thousands of samples were in the reference panel, genotypes for those samples are determined with little error compared to genotypes obtained through imputation.

Response: This is a similar workflow to what has been followed at deCODE genetics for the past several years and described in detail in "Gudbjartsson, D. F. et al. Large-scale whole-genome sequencing of the Icelandic population. *Nat. Genet.* 47, 435 (2015)."

All associations presented are based on the imputed genotypes, even for the case when the individual in question has been genotyped. The reviewer is correct, this can indeed lead to problems with association. In our experience markers that have high genotype quality and high imputation information less commonly suffer from these problems. We have reported the number of markers that fail sequencing batch effects as suggested by the reviewer.

Regarding the association with uric acid, the authors also note that "A second variant rs1189542743, 4Mb downstream at the end of 1p is strongly correlated ($r^2 = 0.68$) and yields a similar association to uric acid." This is an extremely large r^2 for two variants that are so far apart (4 Mb). This is highly unusual. Could this not represent a duplication event, or some other structural re-arrangement? This is the centromere of chromosome 1, so presumably difficult to sequence. How many carriers of each variant were included in the analysis? Was the association with uric acid observed in Europeans? If so, any carriers in non-Europeans?

Response: We have replicated this association in Iceland and added the replication results to the manuscript.

We agree with the reviewer that such high LD over such long physical distance is unusual. However, the relevant measure for LD is the genetic distance between the two markers. According to our genetic map (PMID 30679340) the two markers have genetic coordinates (sex averaged map) of 143.533cM and 143.609cM or a genetic distance of 0.076 cM, i.e. the recombination rate between these markers is approximately 0.02cM/Mb or roughly 60 times smaller than the genomic average of 1.2cM/Mb.

We agree with the reviewer that this long distance could be due to a mapping/assembly bias or a structural variant. To limit the likelihood of mapping bias we search for homozygous carriers of the uric acid locus variants in Icelandic long read sequencing data (PMID 33972781). Two individuals in the long read data were predicted based on our short read data to be homozygous for both variants. These individuals were confirmed as homozygous carriers in the long read data.

Page 11, you state that “A study of WES from 455K individuals in UK biobank recently reported several examples of associations, including those from gene burden analysis. According to the authors, a majority were either in part previously reported, or could not be replicated. It is noteworthy that none of the associations reported here were found in that comprehensive survey of UKB exome variation.” Where exactly in the Backman et al. study is it mentioned that the majority of associations were previously reported or could not be replicated? The abstract says “Of the signals available and powered for replication in an independent cohort, 81% were confirmed”. To me, this indicates that most were replicated. In the text, Backman et al also state that “As it is not possible to exhaustively describe all novel gene associations”, suggesting that many were not previously reported, no? And why is it noteworthy that the handful of associations reported here were not discovered in the Backman et al. analysis? Weren’t the associations that are highlighted in this manuscript of WGS data specifically selected to illustrate examples of associations missed by GWAS and exome-sequencing studies?

Response: Thank you. We have removed this statement.

Reviewer #2

This paper is the much anticipated report on the variant landscape among 150,110 whole genome sequenced samples from the UK biobank. I expect that the findings from this paper will have an immediate impact on the field, in particular the allele frequencies among diverse cohorts and the DR score. It was good to see that they moved beyond the standard small-variant approach and reported on SVs and micro satellites. Overall the paper is an easy to read and straight-forward presentation of the variants. Interest and impact could be improved by deeper discussions of some of the more novel contributions including DR score and non-coding SVs.

Response: Thank you.

I do have issues and questions regarding the DR score. This section is of high interest to the medical genetics and bioinformatics community. An expanded section that better connects DR to health or other traits would be very useful. The authors could simply follow the basic structure in the gnomAD papers (e.g., Karczewski 2020) with updates based on the larger sample size.

Response: We have analyzed the DR in the context of the LOEUF metric from the gnomAD consortium (see detailed answer to reviewer 1), we find correlation between the two metrics, connecting our genome-wide DR to prior work on the coding part of the genome. Our DR metric paves the way for researchers exploring the non-coding genome, and as more of the UKbiobank is sequenced it will gain resolution to detect negative selection in smaller windows.

The methods section cites prior work (J Di et al. 2018) as the basic process for DR. In that paper the authors have their own scoring system called context-dependent tolerance score (CDTS). What is the difference between DR and CDTS? How do they compare?

Response:

The methodologies of CDTS and DR are similar except in the submitted version trimers were used for DR rather than heptamers, in the current version we are using heptamers. This did not have any notable effect on our results as we are using windows of 500bp there is high correlation between the expectation using trimers and heptamers.

We added a comparison of DR score to CDTS as well as measures of functional impact, Eigen, LINSIGHT and CADD. We do see that CDTS and DR are correlated and in fact all the methods are correlated. We note that since Eigen, LINSIGHT and CADD all use GERP as input to their annotations the correlations between those methods and with GERP is higher than for DR and CDTS, which rely on a similar methodology but independent datasets. GERP and similar metrics have inherent limitations to measure human specific conservation as it relies on regions being constrained across mammals (see detailed answer to reviewer 1).

What is the resolution of DR? Coding exons have a mean DR of 28.3. We know from gnomAD (Karczewski 2020) that there is a high degree of variability in the constraint between genes. Haploinsufficient genes are more constrained than olfactory genes and essential genes are more constrained than non-essential genes. Is DR granular enough to capture these differences? Does DR agree with LOUEF (gnomAD's score)? If haploinsufficient genes do have a low DR, are there non-coding regions that have as low or lower DR? What regions have the lowest DR?

Response: DR is computed within 500 bp windows to get a reasonable number of sequence variants per window. We have now added DR score computed over 500bp windows at a sliding distance of 50bp.

We have added comparison of DR to the LOUEF quantiles. DR is correlated with LOUEF (Fig. 1h), see detailed answer to reviewer 1, where we describe the dependency between LOUEF, DR and GERP in detail.

In addition to the Gnomad score we fetched lists of genes from the Gene discovery informatics toolkit which provides various constraint metrics, such as the inheritance pattern of genes in OMIM, hand curated list of genes that are lethal in humans, lethality derived from mouse knockout experiments and cell lethality derived from CRISPR cell assay experiments and we correlated those with DR. LOEUF is correlated with these metrics see Figure 3 from (PMID 32461654). The results from this analysis are provided in Table S8, we find that DR is negatively correlated with autosomal dominant inheritance pattern of genes in OMIM but not with an autosomal recessive pattern.

Surprisingly DR is positively correlated with a list of cell essential genes. We find that DR is negatively correlated with a hand curated list of human lethal genes, and genes that are intolerant to LOF in mice.

Our LOUEF and Gene discovery toolkit analysis demonstrate that the DR is informative about haploinsufficient genes, larger sequencing projects or other analytical approaches are probably needed to rank recessive genes. DR is agnostic to the gene annotation, despite this we find a correlation with various metrics of gene conservation within coding exons, therefore we believe our DR metric is extremely useful for gauging purifying selection across the whole genome.

We have added overrepresentation analysis with ENCODE regions and see that regions annotated by ENCODE as candidate cis-regulatory elements (cCREs) are more likely than expected by chance to be found in DR-low regions. Notably cCREs located in close proximity to transcription start sites, i.e. proximal enhancer-like and promoter-like sequences (pELS and PLS, respectively), are more enriched among DR-low than distal enhancer-like sequences (dELSs).

The result where DR and GERP do not agree is yet another area that could be explored to increase interest and impact.

Response: We agree that these regions are of great interest and these regions were not sufficiently emphasized in our previous submission. We have added extra analysis exploring these regions. We regressed the LOEUF score from the gnomAD consortium, against GERP score and extracted residuals from that fit to construct a human specific LOEUF score (LOEUF|GERP; see LOEUF-DR response to reviewer 1 for more details). As LOEUF is to measure selection in humans and GERP to measure selections in all mammals we expect LOEUF|GERP to measure human specific selection. We find significant correlation between the LOEUF|GERP and DR demonstrating the potential of our approach.

Do the DR scores differ when calculated using the different sub-cohorts? We already know that polygenic risk scores are ancestry-specific, but to my knowledge there has not been an exploration of a similar effect in constraint.

Response: Ancestry impacts the number of sequence variants carried by the sequenced individual when comparing to the reference genome (Fig. 3a). The reviewer raises an important question. The DR rank is based on counts of sequence variants in 500bp windows, therefore the resolution will be limited when restricting to cohorts with a limited number of sequenced individuals.

We have added DR scores computed from the individual cohorts. Our results suggest that the combined set of all cohorts has the highest power to detect human specific conservation, as measured by a decrease in DR score overlapping coding exons. We do see a difference between the cohorts, the XBI cohort is by far the largest cohort we present and XAF and XSA are much smaller and of similar size (Fig S11). Despite this XAF shows a greater decrease in coding exons than the two other cohorts when aggregated across all coding genes. We believe that this is due to the greater genetic diversity of the XAF cohort, which provides more power to detect depletion.

It is not clear from the data availability section that the DR scores will be made public.

Response: The DR score will be made public along with the final version of the manuscript, we added a statement to the data availability section.

The SV section gives specific connections between traits and variants that disrupt exons, which is great. An outstanding question in the SV field is what to do with non-coding SVs. The authors discussed associations between non-coding SNVs and traits (e.g., height). From the stats associated with the SVs discussed it seems that the authors did (or could do) a similar trait association using SVs. Were there any associations between non-coding SVs and traits? Even if this answer is no, it would be very interesting.

Response: We have already presented four associations of SVs and phenotypes, two of which are non-coding. The manuscript submitted was not meant to present all associations that can be obtained from the dataset, rather we sought to highlight a number of applications of the large dataset. We have recently used this dataset to identify a number of non-coding structural variants associating with hematologic traits

<https://www.medrxiv.org/content/10.1101/2021.12.16.21267871v1>.

Referee #3 (Remarks to the Author):

In this article the authors are analyzing the first release of whole genome sequencing (WGS) data from the UK Biobank, representing ~150k individuals in total. Throughout the manuscript the authors draw direct comparisons to what WGS data allows them to discover and analyze in comparison to whole exome sequencing (WES) data and genotype SNP data. In particular the authors look at the genome-wide rare variation they now have access to, they create a new metric to look at potential evidence for purifying selection/functional constraint (Depletion Rank), they compare European and non-European ancestry samples, and they emphasize the importance of both structural variants and microsatellites in gene discovery. This is clearly a unique and exciting dataset as it is the largest collection of WGS data to date. And the authors do discuss what this data brings to the table that has been unavailable to human geneticists up until this point. However, I feel there are multiple areas where

this manuscript could be further strengthened. This includes better explaining some of the points and analyses already shown as well as going deeper in some of the sections. I outline both my major and minor feedback below.

Major Points:

a) Near the end of page 4, the authors discuss the qualities of rare variation and de-novo mutations in their data. While this paragraph provides interesting results, I feel it could use some more context for a broader audience. What is the exact importance of trying to identify the relative rate of

recurrent mutations versus IBD? The authors state “Due to the scale of the UKB WGS data, an observation of the same allele in unrelated individuals does not always imply identity by descent.” This is useful, but I think they should include an additional sentence or two explaining how this distinction is important for the type of analyses we do in human and population genetics. Additionally for this paragraph, the authors show: “Out of the 194,687 Icelandic DNMs we find 53,859 (27.7%) in the UKB set...”; it is unclear to me how to interpret this result. Is an overlap of 27.7% expected? If the authors could provide some additional context, whether analyses, simulations, or just perspective, I think that would also be helpful for the audience here.

Response: We have added the explanation “Inference of haplotypes and imputation typically involves identifying variants that are shared due to a common ancestor - are identical by descent.”

27.7% is considerably higher than the 7.0% fraction of all possible SNPs that we observe in UKB, so this result represents a considerable overrepresentation that can in large part be explained by difference in mutation rate between mutation classes.

b) In the middle of page 5, I find the Depletion Rank (DR) analysis to be particularly interesting. I do have a few questions however. One is, at the end of the section, the authors conclude that DR will help with “...identifying non-coding sequence of functional importance.” Is there a way for the authors to more directly show that this is the case? Do they find enrichment for any type of regulatory architecture amongst the regions that have low DR's? I think since there are tools and methods available now for combining functional genomic information with GWAS hits, trying to run some of these analyses in the regions of low DR could be a way to explicitly show the utility of DR. Specifically, by focusing on regions of low DR, you might alleviate the multiple testing burden and have a more liberal Bonferroni-corrected p-value threshold, thus allowing you to designate more regions as significant for whatever test or metric you end up using. Alternatively, do any of the novel associations that the authors highlight later in the paper occur in the lowest DR regions? That would be a nice connection to bring up if so.

One other question is whether the authors have looked at other commonly used metrics of purifying selection, like Tajima's D or the ExtRaINSIGHT method from the Dukler et al. paper they cite in this section. The authors did compare their DR metric to GERP values, but as they point out this metric uses multi-species comparisons. There are metrics for identifying purifying selection within species as well. The authors could also compare their metric against genome-wide measures of 'deleteriousness' as well, such as CADD, Eigen, or LINSIGHT.

Lastly, is there any reason why these analyses were not conducted in the non-European ancestry populations? I think that could be potentially very interesting, since you might find evidence of particularly recent purifying selection – or at least regional differences in purifying selection.

Response: We have modified the DR methodology, now computing the expected number of mutations from the occurrence of heptamers as opposed to trimers before. We have substantially updated the results and expanded our analysis of the DR score. As discussed in the revised manuscript and our responses of the other reviewers, we have now added analysis which shows that DR measures human specific selection, demonstrating its usefulness beyond that of GERP. We especially point to our analysis of LOEUF|GERP and of constrained genes discussed above.

We thank the reviewer for suggesting we combine functional annotation with genomic hits, in the submitted version we calculated enrichment with GWAS hits. We have now also noted that this can be used in weighing variants for GWAS analysis. Further in response to reviewer's 1 comments we added a FRV of coding variants to compare against DR, we find that the FRV of the top 1% of DR windows corresponds to FRV of a missense mutation, proving a gauge to the user for non-coding genome. FRV is just one of many summary statistics of the site frequency spectrum, but the basis of these summary statistic approaches such as Tajima's D is to derive the neutral expectation for them in the light of recent population expansion and migration. We believe these comparisons of the DR quantiles to coding variants are more appropriate than trying to derive the neutral expectation for summary statistics.

We have added overrepresentation analysis, comparing DR score with the annotations by the ENCODE project. We find that ENCODEs candidate cis-regulatory elements (cCREs) are more likely than expected by chance to be found in DR-low regions. Notably cCREs located in close proximity to transcription start sites, i.e. proximal enhancer-like and promoter-like sequences (pELS and PLS, respectively), are more enriched among DR-low than distal enhancer-like sequences (dELSs).

We have added comparisons to CADD, GERP, Eigen, CDTS and LINSIGHT. Our analysis shows that CADD, GERP, Eigen and LINSIGHT form a correlation cluster (Table S8), and CDTS and DR form another one. DR and CDTS both use a similar approach, and do not use species conservation as CADD, GERP, Eigen, and LINSIGHT do. This has pros and cons, DR is more geared to detect human specific selection compared to GERP (see detailed answer to reviewer 1), but lacks resolution compared to GERP with the current genomic data.

In addition to our original analysis, we have also performed a cohort specific analysis of DR. Our analysis suggests that using all cohorts together has the most power. Aggregated across all the genes we see there is a higher depletion of variants in the coding exons for the XAF than the XBI (Fig. S11). This is likely due to the greater sequence diversity of XAF individuals. Unfortunately, we are limited in the number of variants per window for the XAF and XSA cohorts to estimate regional differences between the cohorts. Our analysis demonstrates the value to sequence the individuals of underrepresented ancestries.

c) At the top of page 7 in the imputation section, the authors state: “It is thus likely that the UKB reference panel provides the best available option for imputing genotypes into population samples from Africa and South Asia.” This is a statement that can be supported by additional analyses, so I am unsure why they were not conducted. Unless I am missing something, only one reference panel was used. To make a statement about how this reference panel performs for non-European ancestry populations versus other reference panels, why not explicitly conduct imputation runs using some of the other reference panels out there, like the 1000G, CAAPA, or TOPMed reference panels? If this new, UKB WGS-based reference panel was indeed better for certain non-European ancestry populations, that would be an impactful finding. But, as far as I can tell, this comparison has not actually been conducted yet.

Response: We have added comparison of imputation accuracy to a previous imputation by Bycroft et al. imputed using data from the Haplotype Reference Consortium, UK 10k and 1000 genomes project.

Conclusively determining whether this panel is the best possible option among all possible options is a considerable task and outside the scope of this manuscript, we have updated the text so that it now refers to “likely one of the best available”.

d) For both the new associations highlighted with the structural variants and microsatellites, it is certainly interesting and important to emphasize that WGS data allows you to better analyze these classes of variants. However, in each section, I think it would strengthen the association results to do some additional follow-up to further support causality. Is there any evidence of any regulatory architecture being present nearby these variants, such as histone mark, methylation, or expression QTLs? If so it might be possible to conduct some colocalization analyses. Are any of these genes present in OMIM or are previous drug targets? Is there any possibility to do a replication analysis for any of these results, such as what Backman et al. 2021 Nature did with the UKB WES and DiscovEHR data? Could the gnomAD WGS data be useful here?

I think the use of NMRK2 and the right atrium heart RNA-seq data is a great start, but are there any links to a relevant phenotype that can be shown with that data? I’m not sure how impactful it is to use that data just to validate the presence of the deletion; connecting the deletion, its association with age at menarche, and its expression in the heart would certainly be a very interesting finding.

Overall, I think trying to take some of these associations ‘one step’ further in terms of showing causality would be very helpful, whichever direction in particular the authors think would be best. Both the Backman et al. 2021 paper and the Barton et al. 2021 Nature Genetics UKB WES imputation paper may be useful references for comparison.

Response: The main goal of the manuscript is to present a large dataset and then highlight the advantage of using WGS over WES. The manuscript then highlights the use of this very large datasets, including some of the findings. We believe that there are many other findings that can be made with this dataset and we believe that further functional validation is well outside the scope of the paper.

We now note which of the genes considered are present in OMIM or as drug targets. We have focused our attention on very rare markers, which are not good candidates for eQTL studies.

To answer the reviewer for each of the associations presented:

ALB: It is not surprising that a deletion near the ALB gene would lower Albumin levels, the novel finding is that we can identify individuals that carry the deletion and thus have lower Albumin levels. Further we show that the deletion associates with calcium and cholesterol levels.

CACNA1A: The association with Familial hemiplegic migraine (FHM1), Epilepsy, Episodic Ataxia Type 2 (EA2) and Spinocerebellar ataxia type 6 (SCA6) have been previously reported. The advance of this study is that we can directly identify carriers of this microsatellite variant, a variant difficult to genotype from WGS data and further confirm this known association.

DMPK: The association with Myotonic Dystrophy has been previously reported. The advance of this study is that we can directly identify carriers of this microsatellite variant, a variant difficult to genotype from WGS data and further confirm this known association and show the clear increase in risk with additional repeat copy numbers.

GHRH: The fact that a variant near GHRH (growth hormone receptor hormone) is associated with height is not surprising. The novel finding here is that we are able to identify this variant from WGS, which can not be identified with WES, and then show that this variant associates with height.

HBB: The marker presented is known to be clinically important, we demonstrate that it can be imputed into a larger set of individuals.

NMRK2: We already show that the variant does affect the expression of the gene in heart tissue.

PCSK9: This gene is a well known drug target, with two drugs targeting PCSK9 (Repatha (Evolocumab) and Praluent (Alirocumab)). The novel discovery here is that we can identify an individual from the XAF cohort who has this gene knocked out (due to a homozygous LoF variant) and replicate an association with a deletion in this gene and non-HDL cholesterol levels.

PIEZO1: The variant presented has already been reported for hemoglobin concentration, the novel finding is that carriers of the variant can be imputed from this dataset.

TAC3: In addition to the results previously presented we now present a replication in Denmark. Rare coding variants in TAC3 and its receptor TACR3 have been reported in OMIM to cause hypogonadotropic hypogonadism under an autosomal recessive inheritance. We now find a variant that affects age of menopause, a condition related to those reported in OMIM.

GCSH: It is not surprising that a deletion deleting two exons of GCSH (Glycine cleavage system H protein) gene would lower glycine levels. The novel finding is that we can identify individuals that are carriers of this deletion and perform the association.

Uric acid/gout: We now provide a replication of this finding in Iceland.

e) As a consideration for a general, additional area of analyses that could help increase the impact of the manuscript, I am wondering about comparing some of the current results versus what happens when you imitate 'low pass sequencing' on the original data. In other words, purposely downsampling your reads and rerunning your SNP discovery pipeline. There is general interest in low pass WGS as a cheaper alternative for a variety of purposes in clinical genomics, and the UKB WGS data seems like a great opportunity to compare how differences in coverage can affect analytical outcomes. This may be a non-trivial set of additional analyses, but I think further exploring analyses that can only be done with this dataset, such as direct comparison of analyses using different levels of sequence coverage, would be great for this manuscript.

Response: The question raised by the reviewer is outside is very interesting and definitely a worthwhile project. However, given the considerable amount of data generation, data description and diverse analyses already necessarily presented in this manuscript, we consider this task to be outside the scope of our study. We do note that the WGS data (bam files and genotype calls) will be available on the UKB research analysis platform, RAP) for other groups to perform these kinds of analyses. We also note that it would not be advisable to use the same pipeline, unmodified, to any sequence coverage. In particular, a design for a study with very low sequence coverage would use software to more efficiently draw inferences from those data. Our design was to have at least 95% of

the genome covered to at least 15x coverage in each sample. Nearly half of the variants detected in this study are singletons, detected in only one sample, and a large majority of the variants are rare. GraphTyper considers only markers that have support from at least 4 high quality base pairs and two proper pairs supporting them in the same individual. The markers also need to be supported by reads coming from both the forward and reverse strands, are both first- and second-in-pair, and must not all have the same starting position. At 15x coverage the probability that a variant observed in a single individual would be misclassified due to random sampling is 3.5%. We have added this explanation to the supplementary text and also added a histogram of the average sequence coverage across all base pairs of the genome.

Minor Points:

a) At the end of page 3, “As GraphTyper provided the more accurate genotype calls the GraphTyper genotypes were used...”, could some summary performance metric be included to show how much more accurate GraphTyper is? I appreciate there is material in the supplement that is referenced, but I think this is a useful result to support directly in the main text some way.

Response: We have added a brief summary of performance results to the main text.

b) At the beginning of page 4, I think it could be worthwhile to spell out ACMG and also explain what an ‘actionable genotype’ is. Since this is intended for Nature, the audience is going to be more broad than the typical genomics community that reads these papers. I’m not sure if it should be assumed that the audience will exactly understand the point being made in this paragraph. To that point additionally, it could also be worthwhile to include an additional sentence or two explaining the impact of identifying a greater number of ACMG genes. The methods section provides a bit more detail on this analysis. Moving some of that to the main text might help.

Response: Thank you. We have added these explanations.

c) In the middle of page 4, the authors show “...Even inside of coding exons currently curated by Encode, we estimate that 10.7% of variants are missed by WES”. Is this simply poor, uneven coverage in the WES data? Is this entirely rare variants? This seems like a somewhat surprising result for at least the coding exons – I think some further analysis or at least hypothesizing on what is going on here would be appreciated.

Response: This is partially due to coverage and partially due to filters in genotype calling, we have noted this in the manuscript.

d) Near the end of page 5 in the DR section, the authors highlight a correlation analysis between their DR metric and GERP, but the r^2 value of their analysis is only .0049. While technically significant with a p-value of .00052, I wonder how meaningful this result is with an r^2 below .01. Is there a typo here? I'm not sure if it's fair to say that there is indeed correlation between the two metrics when the value is so close to 0, even if significant. Also I believe the authors are citing Figure 4d, not 1d, in this section as well.

Response: Thank you for pointing this apparent discrepancy out. We did notice that the p-value reported was not correct and the p-value is much lower than presented before.

We have substantially increased our analysis of the correlation between GERP and DR. Most of the genome is not conserved and we would not expect that the two scores be correlated in regions that are not conserved. GERP measures conservation among all mammals, while DR measures conservation within humans. The strongest correlation between the two scores is in regions that are highly conserved in all mammals, such as coding exons.

For further clarifications, please see detailed answer to reviewer 1.

e) In the caption of Figure 3, I think it would be useful to explain here how 3b and 3c show that differences in singletons is likely due to the geographic sampling disparities between northern and southern Britain.

Response: We have added the explanation "Differences in singleton counts and number of third relatives are likely a result of denser sampling of individuals living in close proximity to UKB assessment centers." to the caption.

f) Near the top of page 7, the authors state: "We found a number of clinically important variants that can now be imputed from the dataset." Is this statement more emphasizing the importance of the original data or the reference panel created from the data? If it's the latter, has this statement been confirmed by imputing the current dataset with other reference panels? Is it the case that you do not recover most of the same variants or that imputation quality scores are significantly worse? I think some more details here would be useful.

Response: We have added a comparison to the imputation made by Bycroft et al. constructed using data from the Haplotype Reference Consortium, UK10 and 1000 genomes project.

g) The authors can also cite the following, more recent height genetics paper to support their discovery of a large effect height allele: <https://www.nature.com/articles/s41586-020-2302-0>

Response: Thank you, we now cite this paper..

h) In the structural variants section, the authors state the following: “The deletion and rs147068659 are correlated ($r^2 = 0.67$), after conditional analysis the deletion remains significant ($p = 6.4 \times 10^{-8}$) whereas rs147068659 does not ($p = 0.39$), indicating the deletion is causal.” The analyses presented suggest the deletion may be the /lead/ variant for this signal. However, association and conditional analyses do not provide evidence for causality. The use of causal here is inaccurate.

Response: Thank you, we have replaced this with a more accurate statement.

i) For all association analyses conducted, was sequencing site used as a covariate? It is mentioned that roughly two thirds of the samples were sequenced at deCODE Genetics and the rest were sequenced at the Wellcome Trust Sanger Institute. I think at least conducting some sensitivity analyses to check this would be good if that hasn't been done yet.

Response: For the imputed data sequencing center was not added as a covariate. The association results presented are for imputed data.

We have added further analysis of the effect of sequencing center on the association results, a number of markers associate with sequencing center, but these are in mostly markers that have been annotated as being of low quality (not in GraphTyperHQ dataset) or having low imputation information.

Reviewer Reports on the First Revision:

Referees' comments:

Referee #1 (Remarks to the Author):

Thank you for the thorough response, no remaining concerns.

Referee #2 (Remarks to the Author):

On most points the authors have addressed my issues. The exception is the response to the DR granularity. In particular, I still do not know if DR can be used to classify or stratify genes in disease studies. In their response, the authors provided linear regressions for DR across different classes of genes, which is helpful, but does not provide the information needed to know how to interpret a particular DR score. The response included a fine set to consider, I would just like to know what the distribution of DR scores were in those different gene sets. I would also like to know the DR distribution for gene set such as autosomal recessive and olfactory genes to help calibrate the scores in the disease causing sets.

Referee #3 (Remarks to the Author):

This article has been improved by the revisions the authors conducted in response to the reviewers' comments. The expansion of the DR score, the inclusion of validation datasets for some of the association findings, and the editing of some of the language describing the findings are all appreciated. However, there are a few points I still feel should be addressed to help solidify the impact of the manuscript.

Major Comments:

a) The inclusion of the LOEUF is an interesting addition to the DR section of the manuscript. The attempt at creating a version of LOEUF that is human-lineage specific by regressing out GERP scores is also an interesting and creative addition. However, I am unsure if DR as a score is capturing something more human-specific than any other constraint metrics. Could the authors conduct an additional sensitivity analysis by comparing CADD, Eigen, LINSIGHT, and CDTS vs. LOEUF and LOEUF|GERP to show that the DR score is performing particularly well as a human-specific constraint metric vs. other ones? I think if the authors show or explain how DR is a better metric to use than these other constraint metrics, that would add to the manuscript. As another possibility, if the authors could set up the DR section as explaining why this new metric is necessary vs. using other pre-existing metrics, that would be helpful as well. As it stands, the section explains the construction of the DR and links it to the finding of less saturated CpG>TpG mutations; however, it is not clearly stated why one would want to go about creating a new constraint metric to begin with or why we would expect a heptamer-based WGS metric to be better.

b) The authors make a point of highlighting their construction of a European-based cohort (XBI) that includes other ancestries beyond the 'white British' (WB) cohort that is typically used with the UKB

data (eg including Irish ancestry individuals). However, it is unclear from the results how much of what they found was due to their use of this 'expanded' European-based cohort vs. what would have been found using the typical WB cohort. Could the authors redo some of their association analyses using the standard WB cohort to show that there was indeed an advantage to using their expanded XBI cohort? Additionally, are any of the findings in the structural variant and microsatellite sections specific to European-ancestry individuals that are not part of the standard WB cohort? If so, this would be another opportunity to support their highlighting of the XBI cohort.

c) Additionally, were there any structural variant or microsatellite findings in the XAF or XSA cohorts that could be highlighted? Once again, since the diversity of this WGS data is highlighted by the authors, it would be great to see as many examples as possible where that diversity is being utilized.

Minor Comments:

a) On lines 246-248, the authors state, "LOEUF is correlated with genes demonstrating autosomal dominant inheritance¹¹, in line with this we find that DR is correlated with autosomal dominant genes as reported by OMIM40 (Table S8)." Could the correlation be included in the parentheses here?

b) Both at points in the reviewer comments and in Table S9 the authors use Pearson's r when describing the correlation results vs. their use of the linear regression r^2 elsewhere. While ultimately equivalent metrics, I think it might help the authors' presentation of the DR section to use Pearson's r throughout, unless there is a reason I am missing on why each metric is appropriate for different parts of the section.

Author Rebuttals to First Revision:

Referee #2 (Remarks to the Author):

On most points the authors have addressed my issues. The exception is the response to the DR granularity. In particular, I still do not know if DR can be used to classify or stratify genes in disease studies. In their response, the authors provided linear regressions for DR across different classes of genes, which is helpful, but does not provide the information needed to know how to interpret a particular DR score. The response included a fine set to consider, I would just like to know what the distribution of DR scores were in those different gene sets. I would also like to know the DR distribution for gene set such as autosomal recessive and olfactory genes to help calibrate the scores in the disease causing sets.

Response: We thank the reviewer for an interesting suggestion that highlights one of many applications for the DR score. However, we believe that this particular analysis is outside the scope of the manuscript. We note that the focus of the DR score is not on protein coding genes. For the protein coding genes we believe constraint metrics incorporating the codon table are more appropriate, such as LOEUF or pLI from the Gnomad consortium. The DR score has been provided with the current version of the manuscript, such that further stratifications and analyses can be performed by the reader.

We have already provided the correlation of DR score with autosomal recessive genes in Supplementary Table 8. We followed the suggestion of the reviewer and correlated the DR score within genes with GO categories, using linear regression and a categorical variable for presence in the category. These categories showed the strongest correlation as ranked by p-value:

GO	Estimate	pvalue
GO:0004984	18.05	8.86e-57
GO:0050911	18.05	8.86e-57
GO:0005515	-3.92	7.79e-40
GO:0045095	25.91	1.66e-35
GO:0005549	27.86	2.59e-33

The category is GO:0004984 is “olfactory receptor activity”, genes in this category had a 18 point higher DR score than genes outside this category. This is not surprising as olfactory genes are known to accumulate a greater number of mutations than other genes.

Referee #3 (Remarks to the Author):

This article has been improved by the revisions the authors conducted in response to the reviewers' comments. The expansion of the DR score, the inclusion of validation datasets for some of the association findings, and the editing of some of the language describing the findings are all appreciated. However, there are a few points I still feel should be addressed to help solidify the impact of the manuscript.

Major Comments:

a) The inclusion of the LOEUF is an interesting addition to the DR section of the manuscript. The attempt at creating a version of LOEUF that is human-lineage specific by regressing out GERP scores is also an interesting and creative addition. However, I am unsure if DR as a score is capturing something more human-specific than any other constraint metrics. Could the authors conduct an additional sensitivity analysis by comparing CADD, Eigen, LINSIGHT, and CDTS vs. LOEUF and LOEUF|GERP to show that the DR score is performing particularly well as a human-specific constraint metric vs. other ones? I think if the authors show or explain how DR is a better metric to use than these other constraint metrics, that would add to the manuscript. As another possibility, if the authors could set up the DR section as explaining why this new metric is necessary vs. using other pre-existing metrics, that would be helpful as well. As it stands, the section explains the construction of the DR and links it to the finding of less saturated CpG>TpG mutations; however, it is not clearly stated why one would want to go about creating a new constraint metric to begin with or why we would expect a heptamer-based WGS metric to be better.

Response: Thank you for the helpful comments and suggestions. We have edited the section to describe better the motivation behind the DR score and its relationship with the CDTS. The DR approach is similar to the CDTS method but uses a larger sequence data set compared to the original CDTS publication, therefore we are building on previous work rather than creating a new metric from scratch. Our aim was to create a sequence constraint metric based solely on the sequence data to be able to discover regulatory features of the non-coding genome in an unbiased manner.

The reviewer raises a valid point, but we do not believe that we can conclusively show there is a “best” constraint metric, as there are pros and cons of the different approaches. For example, the DR and CDTS approach do not use species conservation as the CADD, Eigen and LINSIGHT metrics

nor the epigenetic data used by Eigen and CADD. The addition of these data most likely increases detection power of functionally important regions, but at the expense of biasing the detection towards elements constrained across mammals and known epigenetic features.

With these caveats in mind, we have performed additional computations suggested by the reviewer and we find that DR has the strongest correlation with the LOEUF|GERP metric ($r = 0.155$ 95%CI:0.139-17; $p < 2.2e-16$) compared to the CADD ($r = -0.056$ 95% CI:-0.072- -0.041; $p = 1.5e-12$) and CDTS metrics ($r = 0.098$ 95% CI:0.082-0.114; $p < 2.2e-16$). The Eigen/LINSIGHT metrics do not report values on the coding exons, thus we could not compare those with the LOEUF | GERP metric. The poor correlation between LOEUF |GERP and the CADD metric is not surprising, because CADD largely relies on sequence conservation between species. In the case of the CDTS comparison, DR is based on a much larger sequence data set, demonstrating the value of large human sequencing efforts in deriving such scores. We expect further improvements to be made to the DR score with the final dataset of the 500k UKB WGS and more refined weighting of variants.

These sequence conservation scores are stepping stones toward the goal of identifying constrained non-coding sequences in the human genome. We realize further evaluation of our DR score and other scores are needed to better understand their properties and implications, to facilitate that we will provide the DR score and the underlying data with the final version of the manuscript.

b) The authors make a point of highlighting their construction of a European-based cohort (XBI) that includes other ancestries beyond the 'white British' (WB) cohort that is typically used with the UKB data (eg including Irish ancestry individuals). However, it is unclear from the results how much of what they found was due to their use of this 'expanded' European-based cohort vs. what would have been found using the typical WB cohort. Could the authors redo some of their association analyses using the standard WB cohort to show that there was indeed an advantage to using their expanded XBI cohort? Additionally, are any of the findings in the structural variant and microsatellite sections specific to European-ancestry individuals that are not part of the standard WB cohort? If so, this would be another opportunity to support their highlighting of the XBI cohort.

Response: Thank you for this suggestion. We are wary of performing the exact experiment suggested by the reviewer, for the following reasons. We note that there is considerable extra work involved in demonstrating definitively that a particular subset of individuals provides better association results than another subset of individuals - that we do not believe is justified in this instance. It is well established in statistics generally, and GWAS studies specifically, that larger sample sizes provide more reliable measurements of association. Thus, it follows (and hardly needs to be demonstrated) that our extended XBI cohort is likely to provide greater power to detect true associations than the smaller WB cohort.

Even though care must be taken when comparing association test results for individual loci - because of sampling variance, we nonetheless performed an experiment similar in nature to the one suggested by the reviewer, where we studied the association signals that have been already reported for height. Height is one of the most studied phenotypes in GWAS and has been shown to be highly pleiotropic. Markers reported to be associated with height in the GWAS catalog can in large part be expected to be true positive associations. We tested this phenotype in both the XBI and WB datasets and then computed the average chi-square values of the GWAS catalog markers. We found an average value of 60.2 in the XBI set and 56.7 in the WB set, consistent with us having a higher power to reject true null hypotheses in the XBI cohort.

We note that this experiment still suffers from ascertainment bias as the UKB is probably one of the most extensively studied cohorts in the world and many of the GWAS catalog markers may have been discovered in the UKB.

c) Additionally, were there any structural variant or microsatellite findings in the XAF or XSA cohorts that could be highlighted? Once again, since the diversity of this WGS data is highlighted by the authors, it would be great to see as many examples as possible where that diversity is being utilized.

Response: Yes, there are such findings that could be highlighted and we have given examples here <https://www.medrxiv.org/content/10.1101/2021.12.16.21267871v1>. We have chosen not to highlight further examples in our manuscript due to length restrictions. As pointed out in the manuscript and in our previous response our goal is not to exhaustively report all associations. We respectfully disagree with the reviewer's assessment that "it would be great to see as many examples as possible". This would make for an excessively lengthy exposition which we do not believe would be of interest to the general reader. We note that the underlying data are provided on the UKB RAP, allowing other scientists to further investigate associations with phenotypes.

Minor Comments:

a) On lines 246-248, the authors state, "LOEUF is correlated with genes demonstrating autosomal dominant inheritance¹¹, in line with this we find that DR is correlated with autosomal dominant genes as reported by OMIM40 (Table S8)." Could the correlation be included in the parentheses here?

Response: Thank you. We have made the suggested edit.

b) Both at points in the reviewer comments and in Table S9 the authors use Pearson's r when describing the correlation results vs. their use of the linear regression r^2 elsewhere. While ultimately equivalent metrics, I think it might help the authors' presentation of the DR section to use Pearson's r throughout, unless there is a reason I am missing on why each metric is appropriate for different parts of the section.

Response: We agree with the reviewer that having consistency is paramount for readability. As in other sections we use r^2 we would like to stick to that for this section as well. We have also added the r values to tables S8 and S9.